# GenSpace: Benchmarking Spatially-Aware Image Generation

**Zehan Wang**[1,2*], **Jiayang Xu**[1*], **Ziang Zhang**[1],

**Tianyu Pang**[3†], **Chao Du**[3], **Hengshuang Zhao**[4], **Zhou Zhao**[1,2‡]

[1]Zhejiang University; [2]Shanghai AI Lab; [3]Sea AI Lab; [4]The University of Hong Kong

https://github.com/SpatialVision/GenSpace

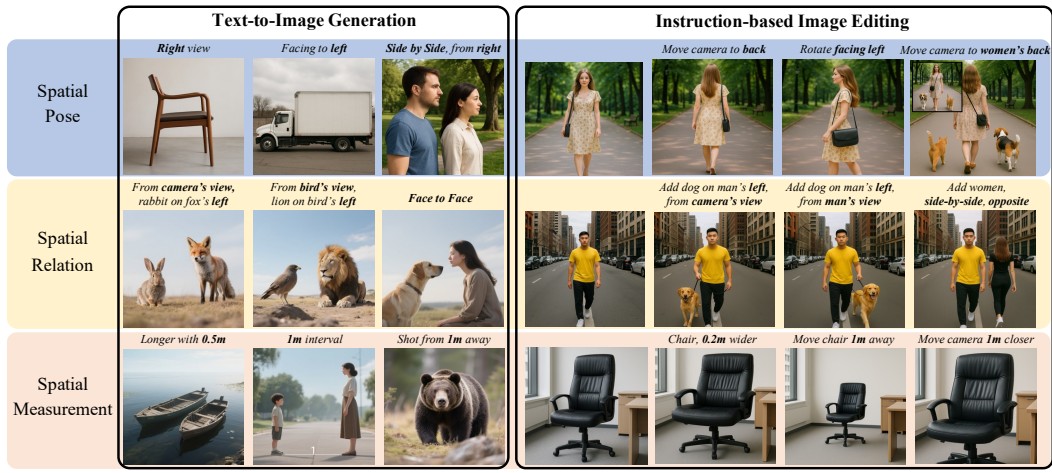

Figure 1: Illustration of **GenSpace**'s three basic evaluation dimensions and nine corresponding sub-domains for text-to-image generation and instruction-based image editing. All results shown are generated by GPT-4o. Zoom in for best viewing.

## Abstract

Humans can intuitively compose and arrange scenes in the 3D space for photography. However, can advanced AI image generators plan scenes with similar 3D spatial awareness when creating images from text or image prompts? We present GenSpace, a novel benchmark and evaluation pipeline to comprehensively assess the spatial awareness of current image generation models. Furthermore, standard evaluations using general Vision-Language Models (VLMs) frequently fail to capture the detailed spatial errors. To handle this challenge, we propose a specialized evaluation pipeline and metric, which reconstructs 3D scene geometry using multiple visual foundation models and provides a more accurate and human-aligned metric of spatial faithfulness. Our findings show that while AI models create visually appealing images and can follow general instructions, they struggle with specific 3D details like object placement, relationships, and measurements. We summarize three core limitations in the spatial perception of current state-of-the-art image generation models: 1) Object Perspective Understanding, 2) Egocentric-

---
[*]Equal Contribution.

[†]Project Leader.

[‡]Corresponding Author.

39th Conference on Neural Information Processing Systems (NeurIPS 2025) Track on Datasets and Benchmarks.

Allocentric Transformation and 3) Metric Measurement Adherence, highlighting possible directions for improving spatial intelligence in image generation.

# 1 Introduction

When photographing, humans often start by thoughtfully arranging both the objects and the camera within the 3D space. Spatial awareness in real-world photography involves imagining the 3D position and orientation of individual objects, and mentally understanding their spatial relationships, quantitatively or qualitatively. For humans, this type of spatial awareness often happens intuitively, allowing us to compose and capture well-structured photographs [11, 1, 68].

On the other hand, image generation models have made remarkable progress in recent years, from diffusion models (Stable Diffusion [49] and FLUX [28]) to the latest unified generative methods (Gemini-2.0-flash [16] and GPT-4o [40]). These models have demonstrated increasingly powerful capabilities in producing realistic and visually appealing images. However, the spatial awareness remains under-explored, despite its crucial role in controllable generation [24, 73, 70], artistic creation [42, 57, 66], and AR/VR applications [14, 38, 4, 67].

To address this gap, we propose **GenSpace**, a new benchmark and evaluation pipeline to assess the spatial awareness of current image generation models. To provide context for our work, we begin by outlining the *Problem Scope* and *Taxonomy* of spatial awareness of image generation:

*Problem Scope*: Aligning with the trend of unified generation models, we comprehensively assess how well these models understand and follow spatial information from text prompts and reference images. Our evaluation covers both text-to-image generation and instruction image editing tasks.

*Taxonomy*: Grounded in the real-world photographic composition process, we systematically categorize spatial awareness capabilities into three dimensions of increasing difficulty:

- *Level 1: Spatial Pose* - Understanding the 3D position and orientation (6 degrees of freedom) of objects and the camera in the scene.

- *Level 2: Spatial Relation* - Reasoning about how objects are positioned relative to each other (spatial layout) and considering different viewpoints (egocentric vs. allocentric).

- *Level 3: Spatial Measurement* - Interpreting quantitative spatial details provides precise controllability, such as object sizes, object intervals, and camera's shooting distance.

Given this problem formulation, a non-trivial challenge arises in how to effectively evaluate the spatial faithfulness of generated images. While previous image generation benchmarks [12, 37, 23, 22] have relied on vision-language models (VLMs) [32, 3, 16, 40] to assess the alignment of generated images with text prompts, recent works [46, 7, 8, 36] show that even the most powerful VLMs still have limitations in spatial reasoning and precise measurement.

To overcome these limitations, we present a novel, automated evaluation pipeline specifically designed to assess spatial faithfulness in images. Our approach utilizes the spatial perception abilities of several visual foundation models (including object detection [34], object segmentation [26, 47], depth estimation [64, 65], orientation estimation [59], and camera intrinsic calibration [74]) to recover the basic spatial pose of each object and reconstruct the 3D scene geometry in the image. We validate our evaluation framework on manually calibrated samples. Our novel evaluation pipeline&metric, leveraging powerful vision foundation models, demonstrates a significant improvement over general-purpose VLMs.

To comprehensively assess model capabilities, we curate 1,800 textual prompts for text-to-image generation, along with 1,800 source image and editing instruction pairs for image editing. We evaluate diverse leading models, including 7 open models and 5 closed models. Our findings reveal that current image generation models often struggle to understand and follow spatial information in text or reference images.

In summary, our contributions are threefold:

- We propose to benchmark spatial awareness capabilities of image generation models, grounded in real-world photography. It covers 3 dimensions and 9 sub-domains of spatial awareness, and considers both text-to-image generation and image editing tasks.

- We develop a novel evaluation pipeline&metric capable of analyzing complex spatial states within generated images. We demonstrate that combining current vision foundation models in this pipeline achieves higher alignment with human spatial perception.

- We evaluate several leading specialized and unified generative models using our benchmark, revealing significant limitations and room for improvement in spatial awareness.

## 2 Related Work

### 2.1 Image Generative Models

With the development of diffusion models [54, 31], Stable Diffusion [49] and SDXL [44], achieve impressive results by utilizing the U-Net diffusion model and extensive pre-trained datasets [51]. Their success inspires lots of subsequent research [6, 19]. Afterwards, progress in transformer architectures [43, 55] and post-training strategies [33, 27, 56] led to bigger and stronger models. Recent methods like SD-3 [10], FLUX [28], HunYuan-DiT [29], and Seedream [13] can generate realistic and aesthetically pleasing images from textual prompts. More recently, the release of GPT-4o [40] has drawn significant attention to unified models, which can process both text and image inputs, naturally unifying generation and editing tasks, demonstrating a leading position in the field.

### 2.2 Benchmarking Image Generation

Initially, image generation models are evaluated using metrics such as Fréchet Inception Distance (FID) [20], Inception Score (IS) [50], and CLIPScore [45]. However, these metrics fall short in assessing complex image-text alignment and subjective attributes. Therefore, more targeted benchmarks and evaluation methods [30, 53, 22, 15] for image generation are proposed. T2I-CompBench [22, 23] and GenEval [15] are object-centric benchmarks, focusing on fundamental object existence, attributes, and quantity, employing object detectors and VLMs for automated scoring. Furthermore, other benchmarks like PhyBench [37], Commonsense-T2I [12], and WISE [39] utilize advanced VLMs [3, 40] to evaluate the understanding of physics and commonsense knowledge.

However, the ability to plan and understand 3D scene layouts—akin to human perception in photography—remains relatively underexplored. Our work aims to address this gap by comprehensively and accurately evaluating the spatial awareness capabilities of current image generative models.

### 2.3 3D Spatial Understanding

Understanding spatial arrangements of images in 3D space is crucial for accurately interpreting complex visual environments. However, recent studies [46, 7, 8, 36, 58, 21] show that even leading general-purpose VLMs struggle with spatial perception and reasoning. To address this, SpatialVLM [7] and SpatialRGPT [8] construct more specialized training data. However, this specialized data still ignores object poses and their allocentric spatial relationships, thereby restricting the models' capabilities.

On the other hand, many benchmarks have emerged to highlight the core challenges of MLLM spatial intelligence. 3DSR-Bench [36] examines issues related to camera viewpoint and object orientation. Thinking in Space [63] investigates spatial memory and reasoning in videos, covering aspects like quantitative distance measurement and egocentric-allocentric transformations. COMFORT [72] explores how different reference systems influence the description of spatial relationships.

## 3 GenSpace

To thoroughly evaluate spatial awareness in image generation, we define three hierarchical dimensions of spatial awareness, from simple to complex (Sec. 3.1). To specifically investigate how models understand spatial information in images and text, and align with the trend towards unified models, we build benchmarks for both text-to-image generation and instruction-based image editing (Sec. 3.2). Finally, we introduce a specialized evaluation pipeline with stronger spatial understanding and reasoning abilities to provide more reliable metrics (Sec. 3.3).

| Sub-domain | Prompt Template |
|---|---|
| Object Pose | "<obj> is facing {*Forward / Backward / Left / Right*} to the viewer." |
| Camera Pose | "{*Front / Back / Left / Right*} view of <obj>" |
| Complex Pose | "<obj1> and <obj2>, side-by-side, shot from <obj1>'s {*Front / Back / Left / Right*}" |
| Egocentric | "From the camera's perspective, <obj1> is {*in Front of / Behind / to the Left of / to the Right of* } <obj2>" |
| Allocentric | "From the <obj2>'s perspective, <obj1> is {*in Front of / Behind / to the Left of / to the Right of* } <obj2>" |
| Intrinsic | "<obj1> and <obj2>, {*Side-by-Side, Same direction / Side-by-Side, Ppposite / Face-to-Face / Back-to-Back*}" |
| Object Size | "Two <obj>, one is {*Bigger / Taller / Longer / Wider*} than another with {*N*} m." |
| Object Distance | "<obj1> separated from <obj2> by {*0.5 / 1.0 / 1.5 / 2.0*} m" |
| Camera Distance | "<obj>, captured from {*1.0 / 2.0 / 3.0 / 4.0*} m" |

Table 1: Prompt templates for text-to-image generation. "<obj>" represents a category name. For each sub-domain, there are 4 distinct templates with different spatial descriptions (i.e., "{*Option 1 / Option 2 / Option 3 / Option 4*}"). The numerical value "{*N*}" is set to a value appropriate for size, height, length, and width. The prompts are simplified for conciseness.

## 3.1 Evaluation Dimension

We focus on three dimensions of spatial awareness (Basic Pose → Qualitative Relation → Quantitative Measurement), grounded in real-world photography. Below, we dive into each dimension and the corresponding sub-domain questions they includes.

### 3.1.1 Spatial Pose

We consider the control of spatial pose, for both objects and the camera, as the most basic spatial concept in image generation. This fundamental capability involves generating objects in a specific orientation or rendering the scene from a particular viewpoint. To evaluate this basic pose control, we use 3 sub-domains of questions:

**Object Pose.** This test checks if the model can generate a single object in the specific pose requested in the text instruction. We focus on single objects to isolatedly assess the basic spatial understanding for various object types without other spatial complexities.

**Camera Pose.** Controlling the camera viewpoint is another basic spatial skill, similar to object pose. For this test, we use the same single-object scenario. We modify the prompt to describe the desired camera viewpoint (e.g., "front view", "right view") instead of the object's orientation.

**Complex Pose.** Controlling the camera in scenes with multiple objects is significantly more challenging than in single-object scenarios. This skill is essential for applications like novel view synthesis. Our complex pose test evaluates a model's ability to envision specific camera views in multi-object scenes while following or preserving the relative positions and relationships between objects.

### 3.1.2 Spatial Relation

Beyond understanding basic object and camera pose, a more advanced challenge in controllable image generation is interpreting the qualitative spatial relationships between multiple objects. Although previous benchmarks [53, 30, 22, 15] have explored similar tasks, they often default to camera-centric perspectives when describing spatial relations. This overlooks the ambiguity of undefined reference systems, causing confusion in real-world descriptions. To reduce this ambiguity and support more use cases, we focus on 3 sub-domains of object relation: Egocentric (Camera-centered), Allocentric (Object-centered), and Intrinsic (View-agnostic)

**Egocentric Relation.** The egocentric descriptions from the viewpoint of the camera or observer are generally intuitive for humans. Previous benchmarks only consider the simple spatial terms like "on the right" (meaning the object is on the right side of the image). However, relying on this unspecific reference system causes ambiguity in practical applications. By explicitly defining the observational viewpoint, we eliminate ambiguity in describing spatial relationships.

**Allocentric Relation.** While egocentric descriptions are common, they don't cover every situation. Spatial relationships are also often described from the viewpoint of another object within the scene (i.e., allocentric viewpoint). An example prompt is "a model leaning against the right door of a car." Our evaluation examines the ability to understand the perspective of different objects and reasons involving the egocentric-allocentric transformation.

| Sub-domain | Instruction Template |
|---|---|
| Object Pose | "Rotate the <obj> to face {*Forward / Backward / Left / Right*} relative to the viewer" |
| Camera Pose | "Show the {*Front / Back / Left / Right*} view of <obj>" |
| Complex Pose | "Move the camera to the {*Front / Back / Left / Right*} of <obj1>" |
| Egocentric | "Add <obj$_{new}$> {*in Front of / Behind / to the Left of / to the Right of*} <obj>, from the camera's perspective" |
| Allocentric | "Add <obj$_{new}$> {*in Front of / Behind / to the Left of / to the Right of*} <obj>, from the <obj>'s perspective" |
| Intrinsic | "Add <obj$_{new}$> near <obj>, {*Side-by-Side, Same direction / Side-by-Side, Opposite / Face-to-Face / Back-to-Back*}" |
| Object Size | "Change the size of <obj>, make it {*Bigger / Taller / Longer / Wider*} by {*N*} m" |
| Object Distance | "Move <obj> 1m {*Forward / Backward / Left / Right*}" |
| Camera Distance | "Change camera distance: move 1m {*Forward / Backward / Left / Right*}" |

Table 2: Instruction templates for instruction-based image editing. We manually curate feasible source images for each instruction to ensure that the edits are meaningful.

**Intrinsic Relation.** There are also spatial descriptions that are independent of any specific viewpoint. These use particular terms to define the intrinsic relationship between objects, such as "side by side" or "back to back." We also assess the model's comprehension of how different objects relate to each other under these view-independent conditions.

### 3.1.3 Spatial Measurement

Advancing this, the generation of images incorporating precise spatial measurements is a highly desirable feature for achieving controllable and spatially-aware image synthesis. Our assessment focuses on the model's proficiency in generating or adjusting images according to 3 fundamental types of spatial measurement: Object Size, Object Distance, and Camera Distance.

**Object Size.** We evaluate the model's ability to understand and control the quantitative 3D size of objects, including their length, width, height, and overall size. Since estimating the real-world size of an object from a single image (monocular) is often ambiguous, we primarily focus on the model's capacity to comprehend relative sizes between objects.

**Object Distance.** We assess the model's understanding and application of specific distances between objects, which allows for more precise control over spatial relationships. This includes creating scenes where objects are a specific distance apart or repositioning objects by a specific amount.

**Camera Distance.** We evaluate the model's ability to understand camera position in terms of distance and to visualize how objects and scenes would appear if captured from different distances. This means accurately showing changes in perspective, visible detail, and the relative size of objects.

### 3.2 Benchmark Construction

With the development of unified image generation models [16, 40], advanced visual generative systems now support mixed image-text inputs, enabling both text-to-image generation and instruction-based image editing within a single framework. To align with this trend and explore the model's spatial awareness over both images and text inputs, we build a benchmark around the key dimensions described above, covering both text-to-image generation and instruction-based image editing tasks.

**Prompt Generation.** First, we create specific prompt templates for testing each sub-domain. (Tab. 1 provides text prompts for text-to-image generation, while Tab. 2 shows instructions for image editing.) Although the generation and editing tasks are different, our prompts and instructions are designed to evaluate similar spatial awareness capabilities across all 9 sub-domains.

**Task1: Text-to-image Generation.** Prompts for this task describe the spatial relationships between objects and the camera. We use 50 common object categories that have distinct orientations (e.g., car, person, and chair). To make our benchmark more diverse and natural, we use LLM to rephrase these templated prompts into more human-like language while keeping the original meaning, for example, "Back view of a fox" to "There is a fox. This image provides a clear view of the rear portion of a fox".

**Task2: Instruction-based Image Editing.** Instructions aim to change the spatial information of objects or the camera in existing images. We manually select source images for each sub-domain from both model-generated images and the internet. The object name in the instruction corresponds to the main object in each image. The instructions are also rewritten by an LLM for naturalness. Finally, humans check all image-instruction pairs to ensure they are clear, correct, and relevant.

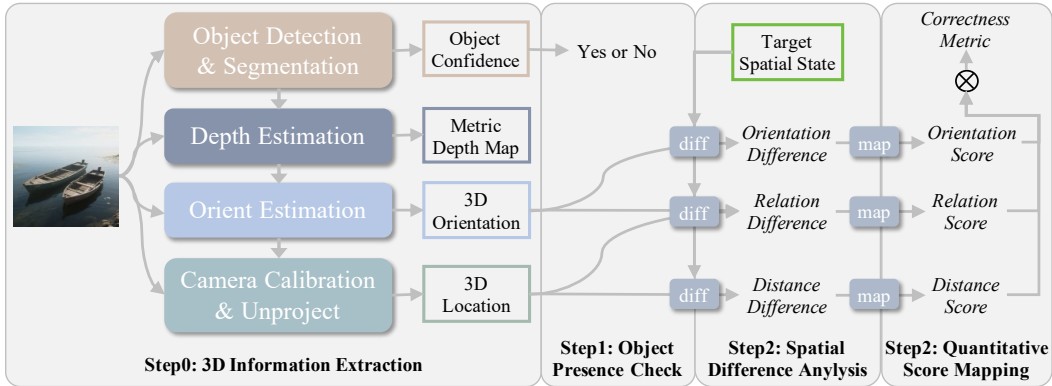

Figure 2: Overview of our evaluation pipeline & metric. We use advanced visual foundation models to extract 3D information from the generated image. We then measure the difference between the estimated spatial state and the target spatial state specified by the prompt or instruction. The differences are converted into a unified score, which serves as the correctness metric.

**Statistic.** Each broader capability dimension is divided into 3 sub-domains. For each sub-domain, we design 4 prompt templates and collect 50 samples per template, yielding 200 samples per sub-domain. Summing up, this results in 1,800 samples for each task, totaling 3,600 samples across two tasks.

## 3.3 Evaluation Pipeline & Metric

Evaluating whether a generated image meets the required spatial information is a non-trivial challenge. As discussed in [46, 7, 8, 36, 60], even state-of-the-art VLM often struggle with spatial understanding and reasoning. To address this, we introduce a Spatial Expertise Pipeline that uses diverse visual foundation models to extract 3D information from a single image and jointly assess the spatial correctness of the generated content.

**Spatial Expertise Pipeline.** For a single image, we use multiple visual foundation models to extract 3D information of the objects: Grounded-SAM [34, 26] for 2D locations, Depth Anything [65] for metric depth, OrientAnything [59] for 3D orientation. We use WildCamera [74] and PerspectiveFields [25] for camera calibration, and then unproject the images into 3D point clouds. In the canonical 3D world, we recognize the absolute 3D location of each object and camera.

**Evaluation Metric.** By using the 3D information extracted, we can effectively analyze the differences between generated spatial states and desired ones. Based on the difference, we determine the correctness of the generated image. Specifically, our evaluation metric is computed in 3 steps:

*Step 1 - Object Presence Check*: First, we need to confirm if the target objects actually appear in the image. For this, we adopt the method from GenEval [15], using object detection to verify the object's presence. To be precise, we use the Grounded-SAM [48] model to detect the object category specified in the prompt within the generated image. If this model fails to detect the target object with its default confidence threshold, the sample is directly marked as a failure and scored as 0.

*Step 2 - Spatial Difference Analysis*: For every textual prompt or editing instruction, we can predefine the intended target spatial state. To measure how much the generated image deviates from this target, we compare the desired spatial state with the 3D information extracted from the synthesized image. This comparison is conducted across three key dimensions: Absolute Orientation Difference (°), Relation Correctness (Yes/No), and Absolute Relative Distance Error (%).

*Step 3 - Quantitative Score Mapping*: To quantitatively assess the overall correctness of an image's spatial state, we map these three types of differences into [0,100] scores. Specifically: For orientation, differences within 30° receive a score of 100. For differences from 30° to 45°, the score linearly decreases to 0. For relation, Yes (correct) scores 100, and No (incorrect) scores 0. For distance, relative errors within 33% are scored as 100. For errors from 33% to 44%, the score decreases to

| Evaluator | Spatial Pose | | | Spatial Relation | | | Spatial Measurement | | | Ave. |
|---|---|---|---|---|---|---|---|---|---|---|
| | Camera | Object | Complex | Ego. | Allo. | Intri. | Size | ObjDist | CamDist | |
| Gemini-2.5-Pro | 58.0 | 56.0 | 54.0 | 76.0 | 50.0 | 48.0 | **67.0** | 47.0 | 52.0 | 56.44 |
| GPT-4o | 49.0 | 45.0 | 54.0 | 83.0 | 42.0 | 46.0 | 49.0 | 58.0 | 53.0 | 53.22 |
| GPT-o3 | 45.0 | 53.0 | 54.0 | 92.0 | 45.0 | 44.0 | 52.0 | 69.0 | 48.0 | 55.78 |
| Ours | **87.0** | **86.0** | **86.0** | **96.0** | **73.0** | **56.0** | 65.0 | **71.0** | **66.0** | **76.22** |

Table 3: Human alignment of different evaluators on spatial understanding, showing their accuracy on manually labeled data. The best results are highlighted in **bold**.

0. Finally, for sub-domain cases that require evaluating multiple conditions simultaneously[4] (e.g., complex poses often depend on both relation correctness and orientation difference), the respective scores are multiplied together as the final score.

# 4 Human Alignment of Metrics

To verify the effectiveness of our evaluation pipeline and metric, we evaluate how different metrics align with human perceptions.

**Testing Data.** For this purpose, three human annotators collect and label 100 generated images for each sub-domain (50 for text-to-image generation and 50 for image editing). Three human annotators label these images using three categories: "Correct", "Partially Correct", and "Incorrect". Furthermore, to ensure label balance, we maintain a roughly similar distribution of these labels within each set of 100 samples per sub-domain. In summary, we have a total of 900 manually annotated samples. We use this test set to evaluate how different metrics align with human judgments.

**Alternative Scoring Methods.** Recent visual generation benchmarks predominantly use advanced VLMs for evaluation. Therefore, we employ three state-of-the-art VLMs, Gemini-2.5-pro [17], GPT-4o [40], and GPT-o3 [41] as comparative baselines. These VLMs are tasked with analyzing the spatial adherence of generated samples to the given text or image-text prompts and assigning a score from 0 to 100 to each sample. To align the fine-grained continuous scores with this categorical system for comparison, we map scores as follows: 0 to "Incorrect," (0, 100) to "Partially Correct," and 100 to "Correct." Finally, we measure how well each method aligns with human perception by comparing its accuracy against manual human labels.

**Results.** Tab. 3 presents the comparative results of different evaluators' alignment with human judgment. Overall, our spatial expertise pipeline and corresponding metric demonstrate a stronger correlation with human recognition. Across sub-domains, our method achieves 76.22% average agreement with manual labels, while the most advanced VLM, Gemini-2.5-Pro, achieves only 56.44%. These comparisons highlight the shortcomings of current VLMs in allocentric perspective reasoning and quantitative spatial measurement, underscoring the necessity of our specialized evaluator.

# 5 Evaluation Results

## 5.1 Experiment Settings

We evaluate 8 models for text-to-image generation: 6 expertise models (SD-1.5 [49], SD-XL [44], DALL-E 3 [5], SD-3.5 [10], FLUX.1-dev [28], and Seedream-3.0 [13]) and 3 unified models (Bagel [9],Gemini-2.0-Flash [16] and GPT-4o [40]). For image editing, we evaluate 6 models: 4 expertise models (InstructPix2Pix [6], ICEdit [71], Step1X-Edit [35], and SeedEdit [52]) and 2 unified models (Gemini-2.0-Flash [16] and GPT-4o [40]). For inference, we employ the official default configuration for each model with fixed random seeds. All experiments are conducted on May 10, 2025.

---

[4]The detailed spatial state conditions required by each sub-domain are provided in the Appendix.

| Model | Spatial Pose | | | Spatial Relation | | | Spatial Measurement | | | Ave. Rank | Arena ELO |
|---|---|---|---|---|---|---|---|---|---|---|---|
| | Camera | Object | Complex | Ego. | Allo. | Intri. | Size | ObjDis | CamDis | | |
| *Expertise Generative Model* | | | | | | | | | | | |
| SD-1.5 | 31.08 | 22.10 | 3.35 | 52.33 | 9.80 | 11.97 | 24.97 | 34.36 | 31.13 | 7.3 | 587 |
| SD-XL | 33.66 | 25.03 | 9.52 | 46.15 | 16.38 | 8.87 | 23.89 | 33.76 | 22.75 | 7.7 | 841 |
| DALL-E 3 | 50.37 | 46.81 | 10.92 | 65.74 | 17.45 | 16.63 | 30.32 | **41.91** | 25.69 | 4.7 | 937 |
| SD-3.5-L | 42.85 | 31.48 | 5.90 | 73.03 | 11.15 | **23.55** | **31.03** | 33.05 | 24.83 | 5.3 | 1028 |
| FLUX.1-dev | 40.42 | 31.11 | 12.28 | 63.39 | 13.17 | 19.40 | 29.16 | 30.72 | 31.98 | 5.4 | 1046 |
| Seedream-3.0 | 53.75 | 61.62 | 13.70 | 84.84 | 18.56 | 17.02 | 26.24 | 30.89 | 26.13 | 4.0 | 1149 |
| *Unified Generative Model* | | | | | | | | | | | |
| Bagel | 43.34 | 46.65 | 13.47 | 72.10 | **22.53** | 19.12 | 30.77 | 36.86 | 29.01 | 3.6 | - |
| Gemini-2.0-Flash | 54.77 | 52.93 | 10.92 | 81.85 | 17.50 | 14.07 | 24.61 | 28.04 | 31.13 | 4.9 | 962 |
| GPT-4o | **59.41** | **62.72** | **25.01** | **94.55** | 21.21 | 19.08 | 30.47 | 41.33 | **35.19** | **1.8** | **1152** |

Table 4: Benchmarking the spatial awareness within text-to-image generation.

| Model | Spatial Pose | | | Spatial Relation | | | Spatial Measurement | | | Ave. Rank |
|---|---|---|---|---|---|---|---|---|---|---|
| | Camera | Object | Complex | Ego. | Allo. | Intri. | Size | ObjDis | CamDis | |
| *Expertise Generative Model* | | | | | | | | | | |
| InstructP2P | 5.02 | 4.49 | 0.00 | 55.71 | **43.36** | 8.44 | 8.33 | 4.09 | 3.96 | 5.9 |
| ICEdit | 4.04 | 5.61 | 0.23 | 63.36 | 42.40 | 12.52 | 9.37 | 5.35 | 5.46 | 4.8 |
| Step-Edit-X | 3.78 | 5.70 | 0.02 | 70.01 | 30.06 | 14.45 | **18.03** | 4.65 | 3.28 | 5.4 |
| SeedEdit | 23.51 | 16.03 | 0.78 | 85.91 | 34.33 | **22.49** | 11.46 | 7.03 | 8.80 | 2.7 |
| *Unified Generative Model* | | | | | | | | | | |
| Bagel | 45.37 | 49.55 | 0.77 | 78.51 | 38.74 | 17.03 | 11.11 | 6.79 | 4.94 | 3.4 |
| Gemini-2.0-Flash | 46.81 | 38.12 | 0.17 | 81.19 | 33.88 | 18.50 | 7.02 | 5.04 | 8.63 | 4.0 |
| GPT-4o | **54.38** | **49.94** | **1.80** | **88.47** | 33.62 | 20.55 | 14.05 | **9.97** | **14.45** | **1.8** |

Table 5: Benchmarking the spatial awareness within instruction-based image editing.

## 5.2 Text-to-Image Generation Results

In Tab. 4, we present the evaluation results for text-to-image generation tasks. Note that, to provide a reference for models' general capabilities, we also report the Arena ELO score [5] in Artificial Analysis [2], a model ranking board maintained by lots of human users. Generally, the model rankings on our benchmark align well with human rankings of their overall capabilities, which supports the reliability of our benchmark. Specifically, we find that unified generative models perform better than dedicated image generation models with similar ELO scores, possibly due to the general cognitive improvements gained from unifying image and text inputs. Furthermore, we also observe that closed-source models (DALL-E 3, Seedream-3.0, Gemini-2.0-Flash, and GPT-4o) still comprehensively outperform their open-source counterparts.

Regarding individual dimensions: 1) For spatial pose, even the best models are only about 60% accurate in understanding basic front, back, left, and right views of objects, and the accuracy drops significantly in complex multi-object scenes. 2) For spatial relations, models handle intuitive egocentric relationships almost perfectly. However, they perform very poorly on egocentric-to-allocentric views transformation or intrinsic relationships understanding. 3) For spatial measurement, almost all models struggle to generate images with specific, quantitative measurements.

## 5.3 Instruction-based Image Editing Results

Tab. 5 includes the evaluation results for instruction-based image editing. Overall, the unified generative model shows similar advantages and limitations across different evaluation dimensions as in the text-to-image generation task, highlighting the connection between our benchmarks for both

---

[5]Arena ELO score in May 10, 2025

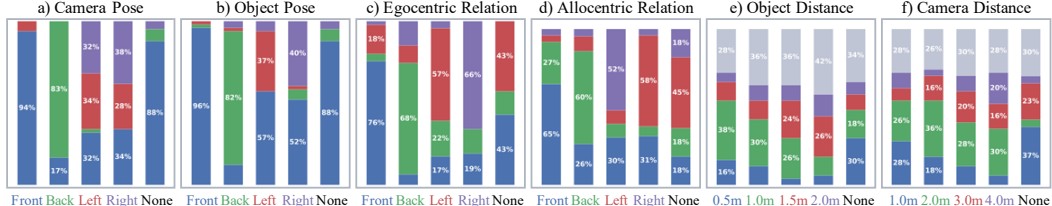

Figure 3: Impact of varying spatial conditions on the spatial states of generated samples. The horizontal axis shows the spatial conditions specified in the prompt, with colored bars representing the resulting spatial states for each condition. Statistics derived from GPT-4o's text-to-image outputs on 6 sub-domains, and analysis of other models in the Appendix.

tasks. Furthermore, GPT-4o performs best across most sub-domains, demonstrating a next level of general generative capability compared to other models, while its absolute spatial reasoning still has significant room for improvement.

Notably, we find that some specialized editing models are almost completely ineffective at modifying existing spatial pose and shape. We infer this is due to the limitations of current open-source editing training data. Most existing instruction-based editing data [6, 61, 69] focuses on simple modifications like changing color, adding/removing objects, or style transferring while strictly keeping the overall image structure. This focus on simplified tasks limits their capability in spatial structure edits. Unified generative models, in contrast, are trained on broader data for both generation and editing, and tend to regenerate images based on instructions rather than merely modifying them.

## 5.4 Core Limitations in SoTA Model

Beyond analyzing the final scores for model strengths and challenging sub-domains, we also dive into detailed error analysis, visualized in Fig. 3. Combined with the overall results of each sub-domain, we summarize the core limitations of current state-of-the-art generative models regarding spatial understanding as follows:

**Limitation in camera location understanding.** Fig. 3 *a* illustrates that generative models struggle to distinguish between side views (e.g., "right/left view") of objects. However, directly stating the wanted orientation in the final image (e.g., "facing right/left") significantly reduces this confusion, despite describing the same spatial state. This suggests that current models favor direct, object-centric descriptions. Their weakness in indirect spatial reasoning and camera position understanding hurts their capacity for complex prompts and spatial control.

**Limitation in egocentric-allocentric transformation.** From Fig. 3 *c* and *d*, we observe that even models like GPT-4o still limited to egocentric thinking for object relationships. When given allocentric prompts, the model often reverses the left/right condition. This stems from models' tendency to generate objects facing the viewer, which inverts the object's left/right relative to the viewer's. Thus, despite providing explicit object-centric instructions, models still default to a simple egocentric (image-based) understanding.

**Limitation in understanding metric measurement.** Tab. 4 and 5 show that current models are largely unable to understand or adhere to quantitative 3D spatial measurements. This limitation is even more apparent in Fig. 3 *e* and *f*. Specifying different measurements has little effect on the final generated results. Considering the wide applicability of precisely and quantitatively controlling the image layout, quantitative spatial awareness is a crucial area for improvement.

## 6 Conclusion

In this work, we introduce GenSpace to assess whether current rapidly developing image generative models can control spatial layout in images as human photographers. To this end, we curate benchmarks for spatial awareness over three core dimensions under both text-only and text-image-mixed input. Moreover, we propose a more rigorous evaluation pipeline and metric specifically

for spatial understanding. In human studies, our proposed metric exhibits stronger alignment with human perceptions compared to advanced VLMs. After benchmarking several representative image generative models, we find that the unified generation model GPT-4o, while performing best overall, still shows significant limitations in understanding 1) camera location, 2) perspective transformations, and 3) metric measurements. We hope our empirical findings and insights can guide future research towards achieving stronger spatial control in AI image generation.

**Limitation and Future Work.** We are continuously adding results from more image generative models to GenSpace. We will integrate more advanced visual foundation models when they are released, aiming to make our evaluation pipeline more robust.

## Acknowledgements

This work was supported in part by National Key R&D Program of China (No. 2022ZD0162000) and National Natural Science Foundation of China (No. 62222211, U24A20326, 624B2128, 62422606 and 62201484)

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

## A Detailed scoring criteria for each sub-domain.

In Tab.6, we show the demand for spatial difference for different subdomains.

### A.1 Spatial Pose

For the two subdomains Camera Pose and Object Pose [62, 18], there is only one object in the image, and the criterion for judging is the orientation of the object, so there is only one difference that needs to be introduced, orientation difference. For Complex Pose, which involves multiple objects, it is necessary to maintain the side-by-side and same orientation relationship of these objects in the text-to-image generation task; while in the image editing task, it is necessary to maintain the general spatial relationship between the objects before and after editing without any major changes. Therefore, in addition to orientation difference, it is necessary to introduce relation difference.

### A.2 Spatial Relation

All three subdomains of this dimension encompass multiple objects. Both orientation and relation differences are required in the scoring stage.

### A.3 Spatial Measurement

In this domain, Distance Difference is necessary in order to measure the length/width/height/volume of an object. Besides, Orientation difference is required for the subdomain of Object Size, because we consider the measure parallel to the front-back direction of an object as length, the measure parallel to the left-right direction of an object as width, and the measure parallel to the top-bottom direction of an object as height. Therefore, for the scoring of Object Size, we need to distinguish the length, width and height of an object by its orientation.

| Domain | Sub-domain | Orientation Diff. | Relation Diff. | Distance Diff. |
|---|---|:---:|:---:|:---:|
| **Spatial Pose** | Camera Pose | ✓ | ✗ | ✗ |
| | Object Pose | ✓ | ✗ | ✗ |
| | Complex Pose | ✓ | ✓ | ✗ |
| **Spatial Relation** | Egocentric | ✓ | ✓ | ✗ |
| | Allocentric | ✓ | ✓ | ✗ |
| | Intrinsic | ✓ | ✓ | ✗ |
| **Spatial Measurement** | Object Size | ✓ | ✗ | ✓ |
| | Object Distance | ✗ | ✗ | ✓ |
| | Camera Distance | ✗ | ✗ | ✓ |

Table 6: Difference required for each sub-domain

## B Impact of Spatial Conditions for More Models

**Limitation in camera location understanding.** As shown in sub-Fig.a and sub-Fig.b of Figs.4 to 10, most models also suffer from a lack of understanding of distinguishing between side views (e.g. "right/left view") of objects.

**Limitation in object location understanding.** In addition, the FLUX.1-dev, SD-XL, SD-1.5, and SD-3.5-L show a very clear shortcoming in understanding object orientation (both from the camera pose and the object pose). In most cases they just directly draw the object in front view, no matter what the prompt is.

**Limitation in egocentric-allocentric transformation.** The sub-Fig.c and sub-Fig.d of Figs.4 to 10 illustrate that the other models are almost exclusively limited to egocentric thinking for object relationships, similar to the GPT-4o.

**Limitation in understanding metric measurement.** As with GPT-4o, the other existing models, almost all of them, are incapable of understanding information from quantitative spatial measurements. In particular, the 2m in Object Distance and the 4m in Camera Distance are almost rarely generated by the models, even if prompt tells them to do so, as show in the sub-Fig.e and sub-Fig.f of Figs.4 to 10.

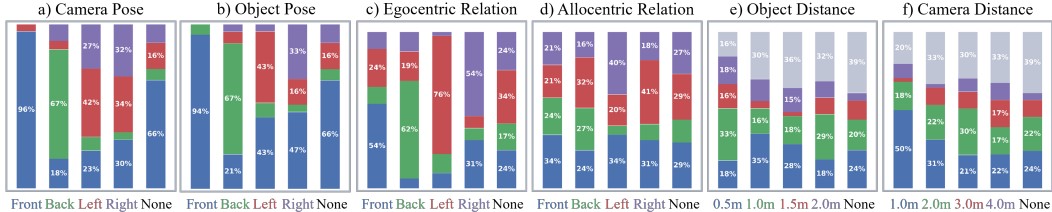

Figure 4: Impact of varying spatial conditions on the spatial states of generated samples from **Gemini-2.5-Pro**.

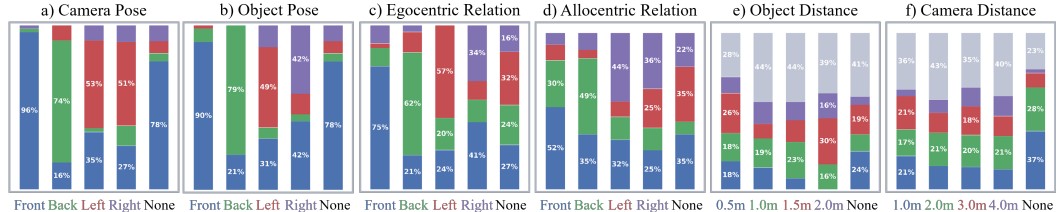

Figure 5: Impact of varying spatial conditions on the spatial states of generated samples from **Seedream-3.0**.

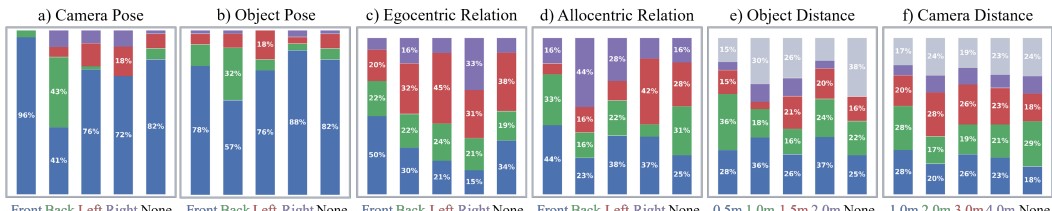

Figure 6: Impact of varying spatial conditions on the spatial states of generated samples from **FLUX.1-dev**.

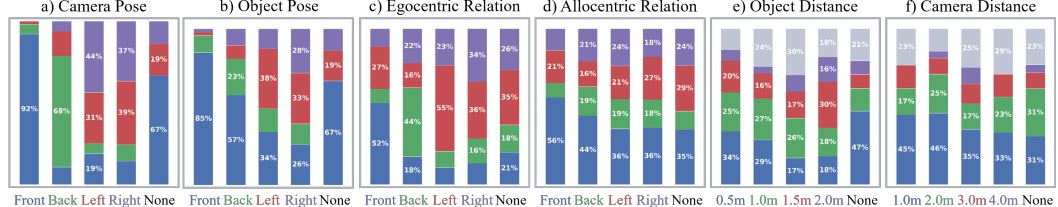

Figure 7: Impact of varying spatial conditions on the spatial states of generated samples from **DALL-E 3**.

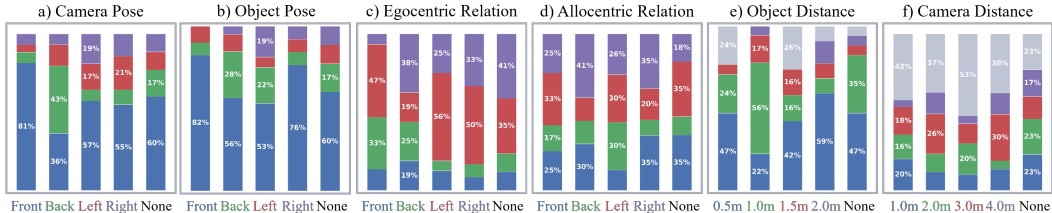

Figure 8: Impact of varying spatial conditions on the spatial states of generated samples from **SD-XL**.

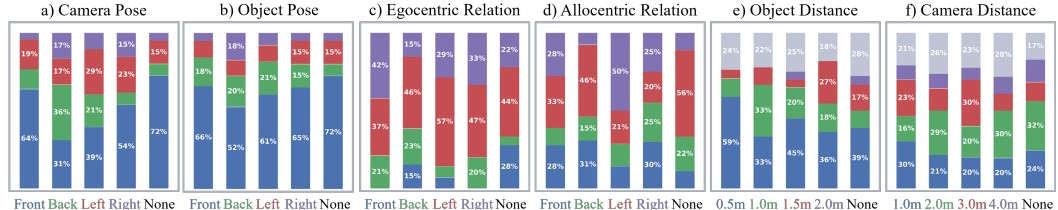

Figure 9: Impact of varying spatial conditions on the spatial states of generated samples from **SD-1.5**.

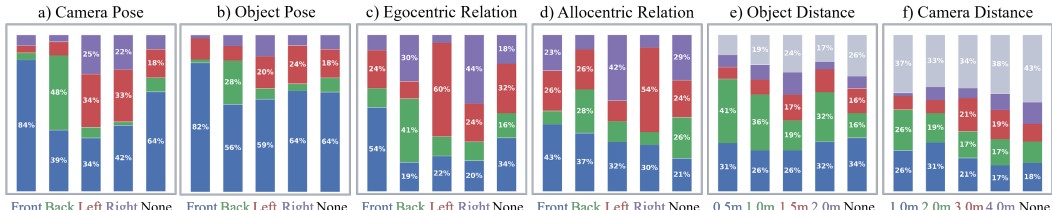

Figure 10: Impact of varying spatial conditions on the spatial states of generated samples from **SD-3.5-L**.

# C  Visualization of Text-to-image Generation Benchmark

In this section, we show the 36 small generation tasks (each subdomain contains 4 tasks) that we covered in the Text-to-image Generation Benchmark. Each task contains eight images generated by eight models prompted by the same instruction. The images that match the instruction are labeled with green boxes, those that do not match are labeled with red models, and those that are partially correct are labeled with yellow boxes.

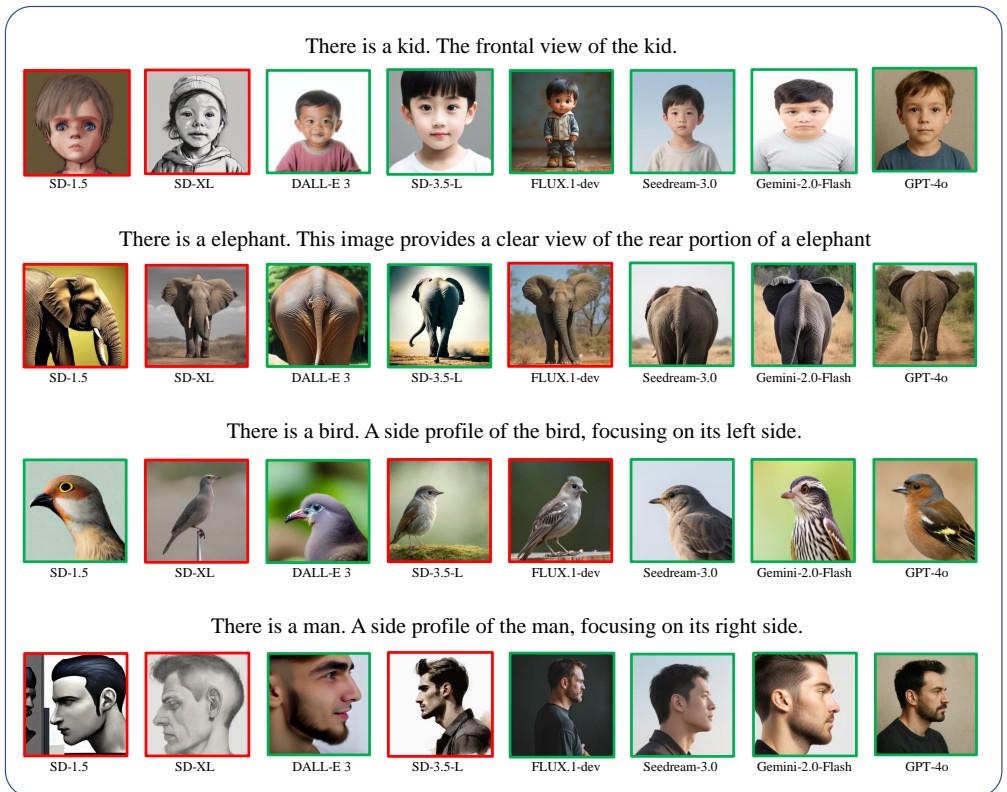

Figure 11: Visualization of Text-to-image Generation Benchmark on the subdomain **Camera Pose**

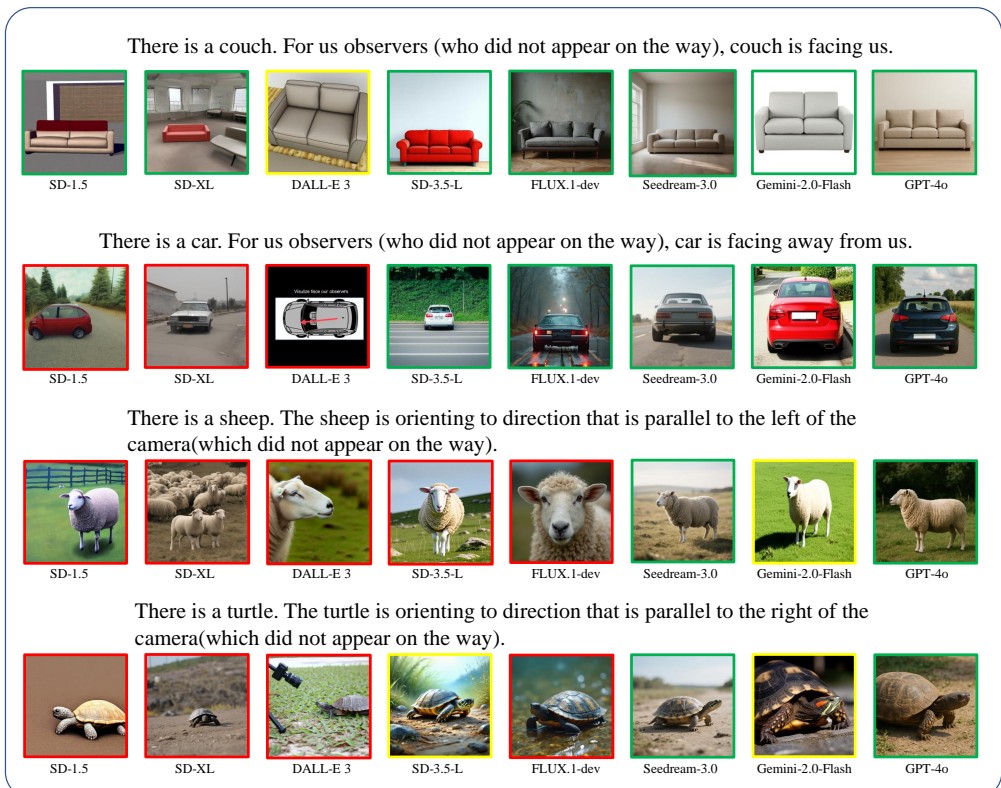

Figure 12: Visualization of Text-to-image Generation Benchmark on the subdomain **Object Pose**

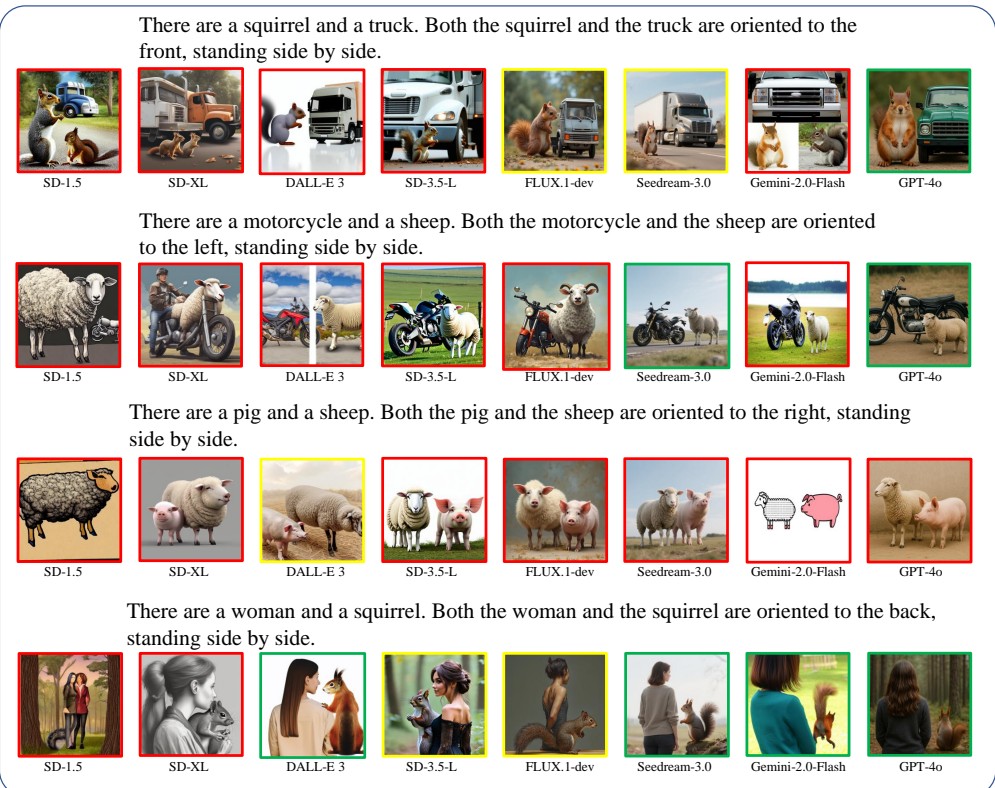

Figure 13: Visualization of Text-to-image Generation Benchmark on the subdomain **Complex Pose**

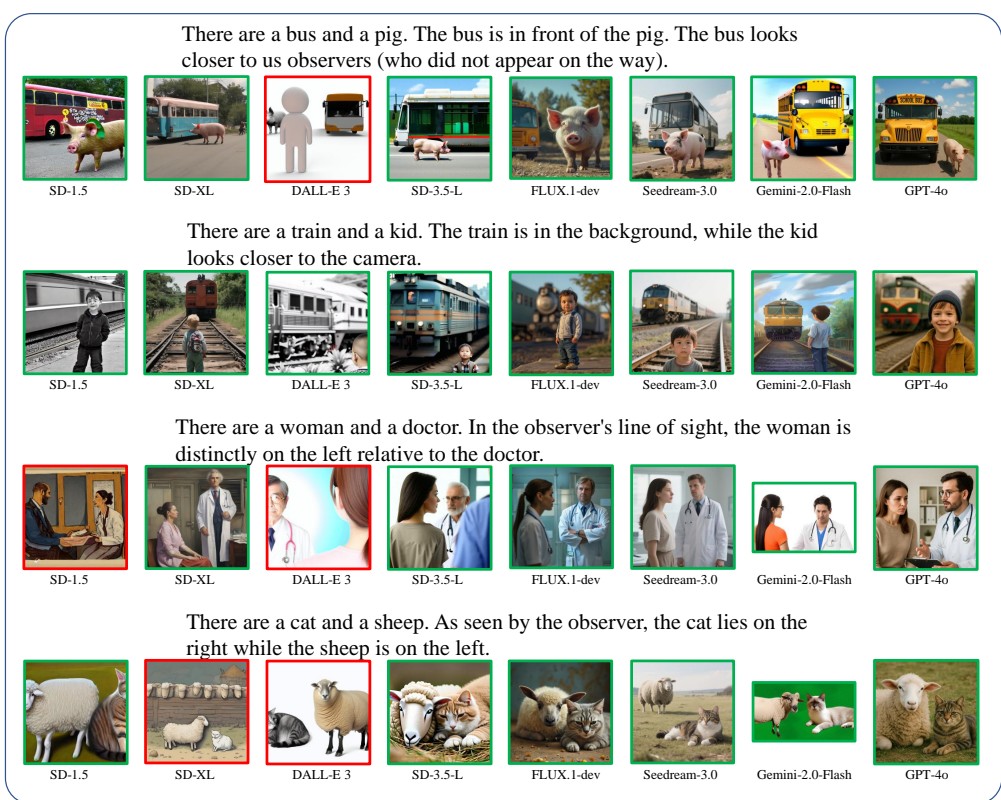

Figure 14: Visualization of Text-to-image Generation Benchmark on the subdomain **Egocentric Relation**

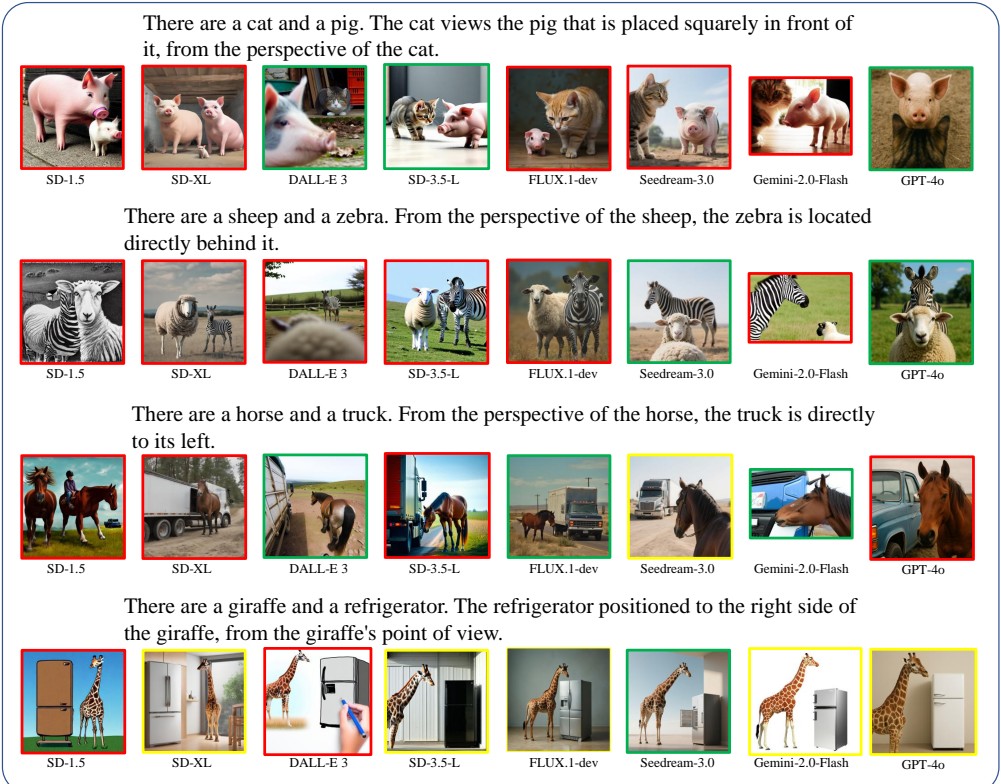

Figure 15: Visualization of Text-to-image Generation Benchmark on the subdomain **Allocentric Relation**

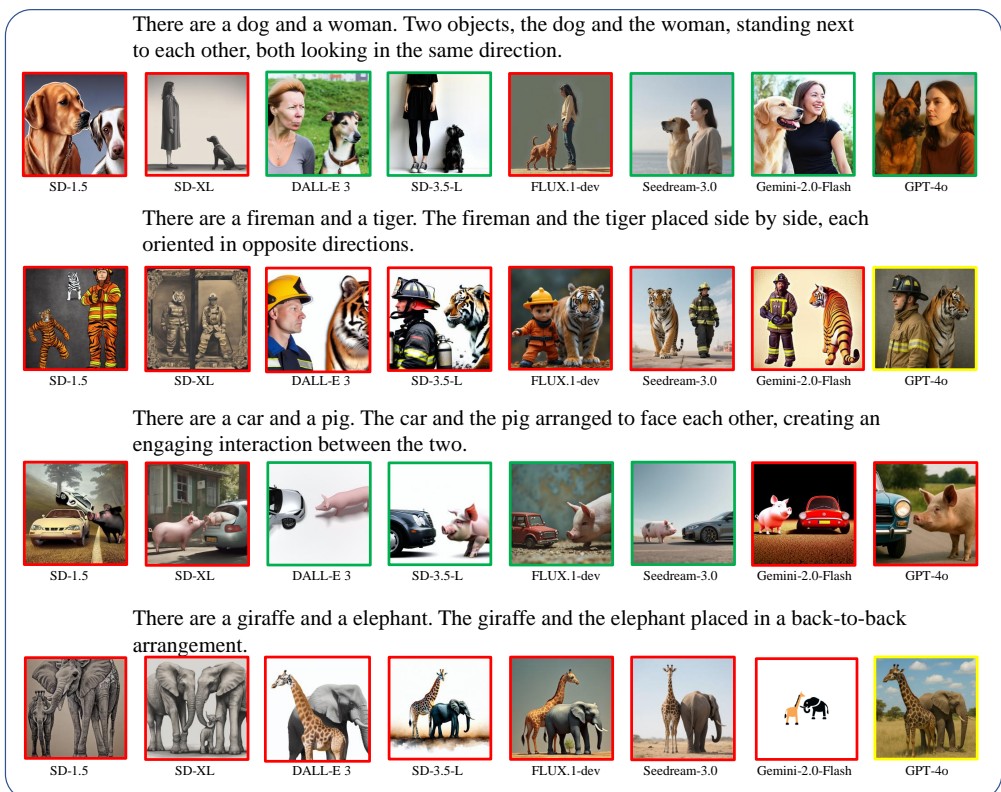

Figure 16: Visualization of Text-to-image Generation Benchmark on the subdomain **Intrinsic Relation**

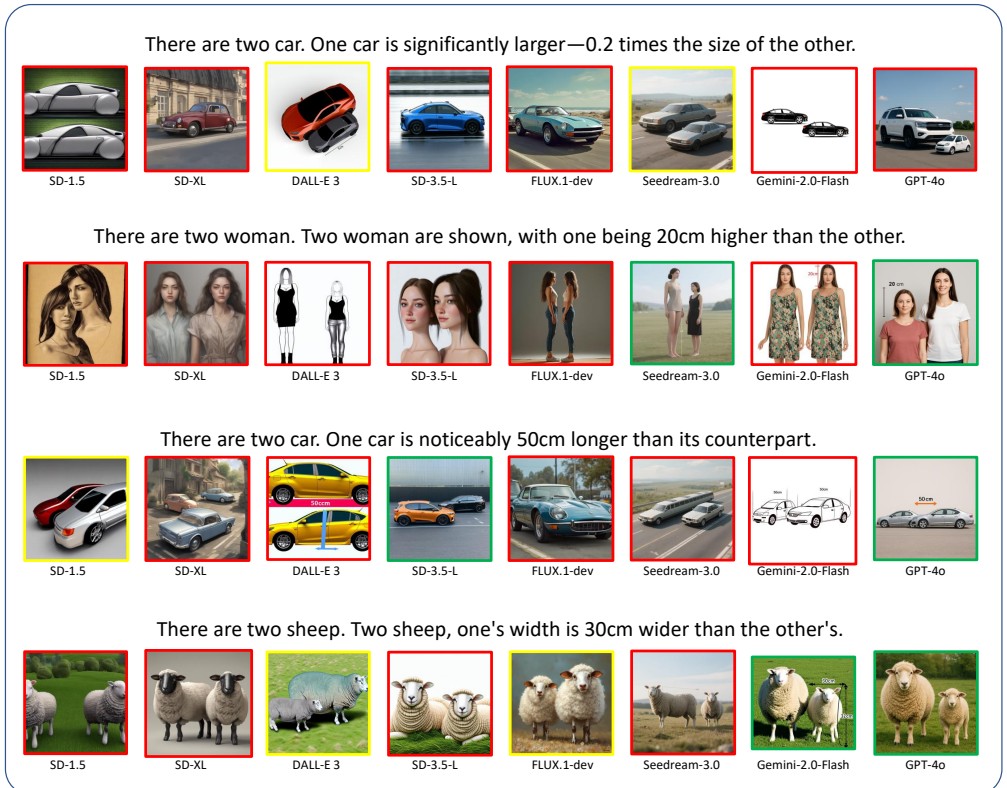

Figure 17: Visualization of Text-to-image Generation Benchmark on the subdomain **Object Size**

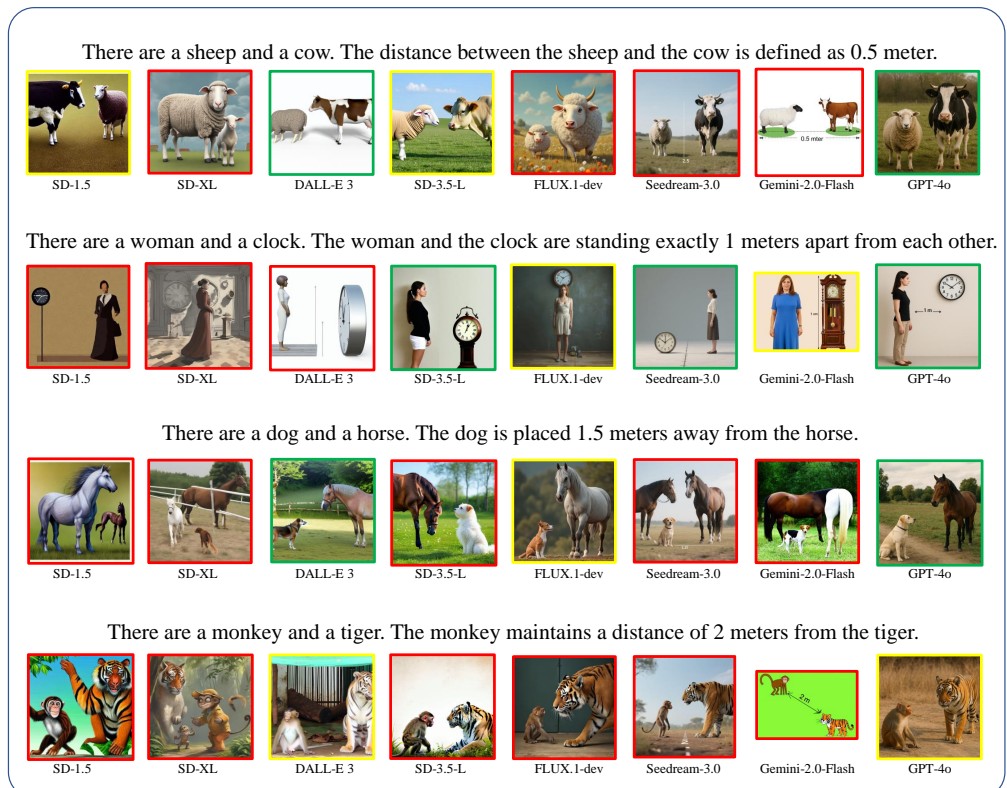

Figure 18: Visualization of Text-to-image Generation Benchmark on the subdomain **Object Distance**

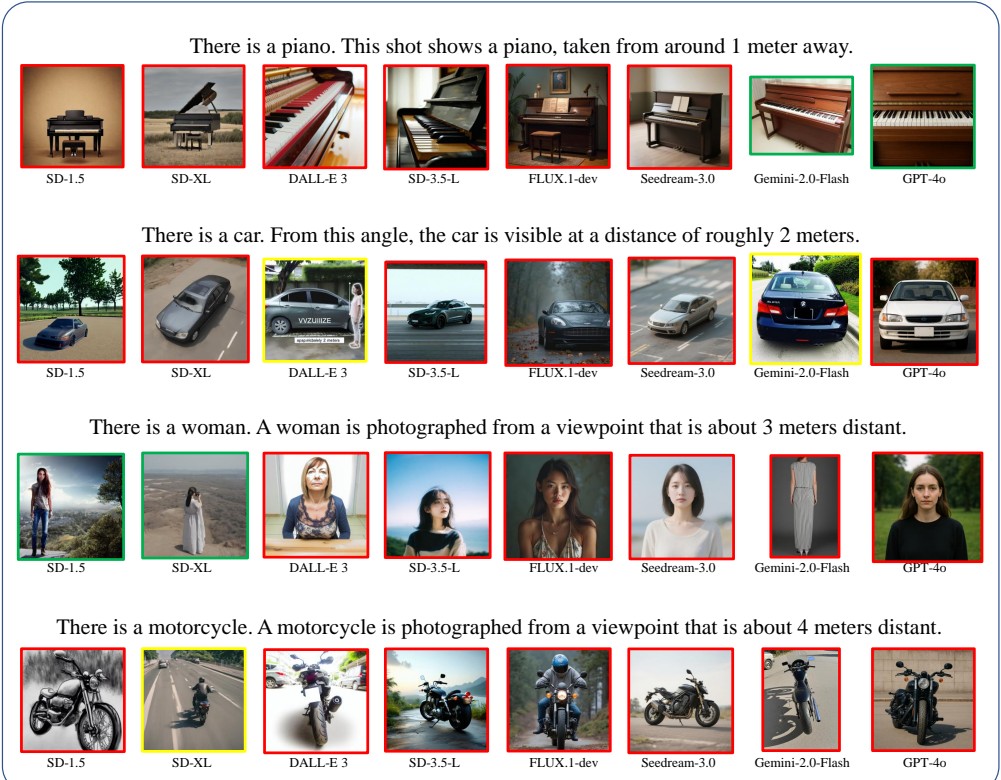

Figure 19: Visualization of Text-to-image Generation Benchmark on the subdomain **Camera Distance**

# D   Visualization of Instruction-based Image Editing Benchmark

In this section, we show the 36 small edit tasks (each subdomain contains 4 tasks) that we covered in the Instruction-based Image Editing Benchmark. Each task contained one original image, and six images edited by six models from the same text prompt. The images that match the instruction are labeled with green boxes, those that do not match are labeled with red models, and those that are partially correct are labeled with yellow boxes.

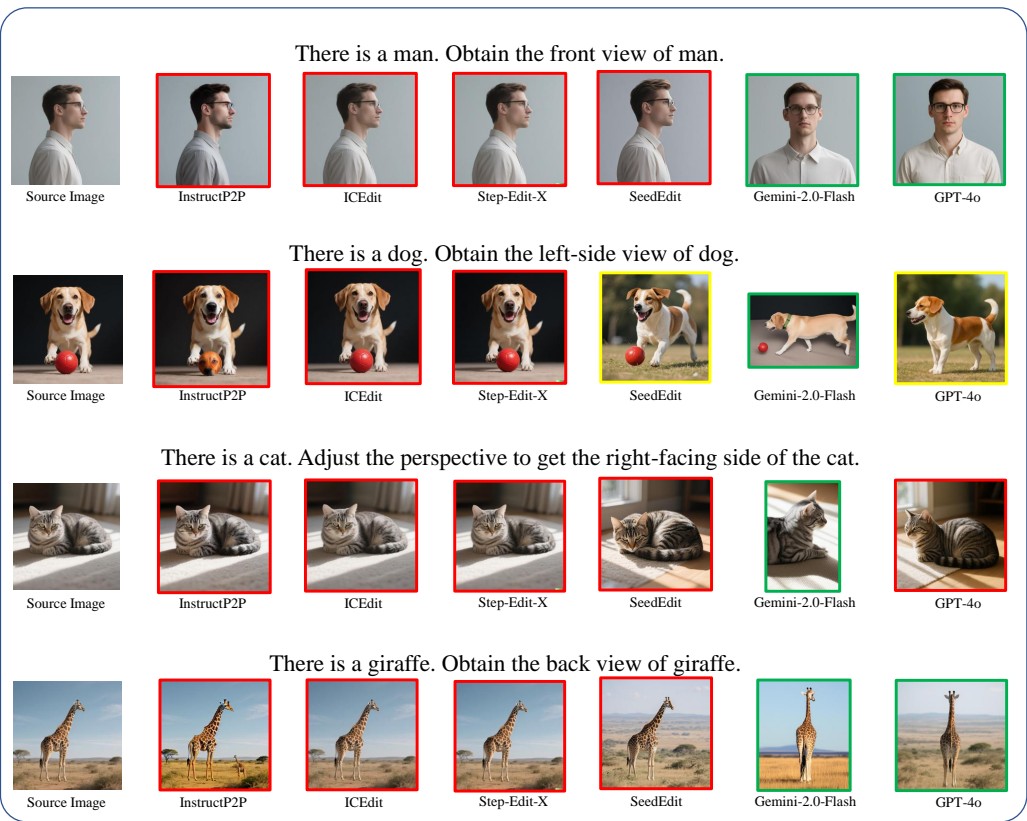

Figure 20: Visualization of Instruction-based Image Editing Benchmark on the subdomain **Camera Pose**

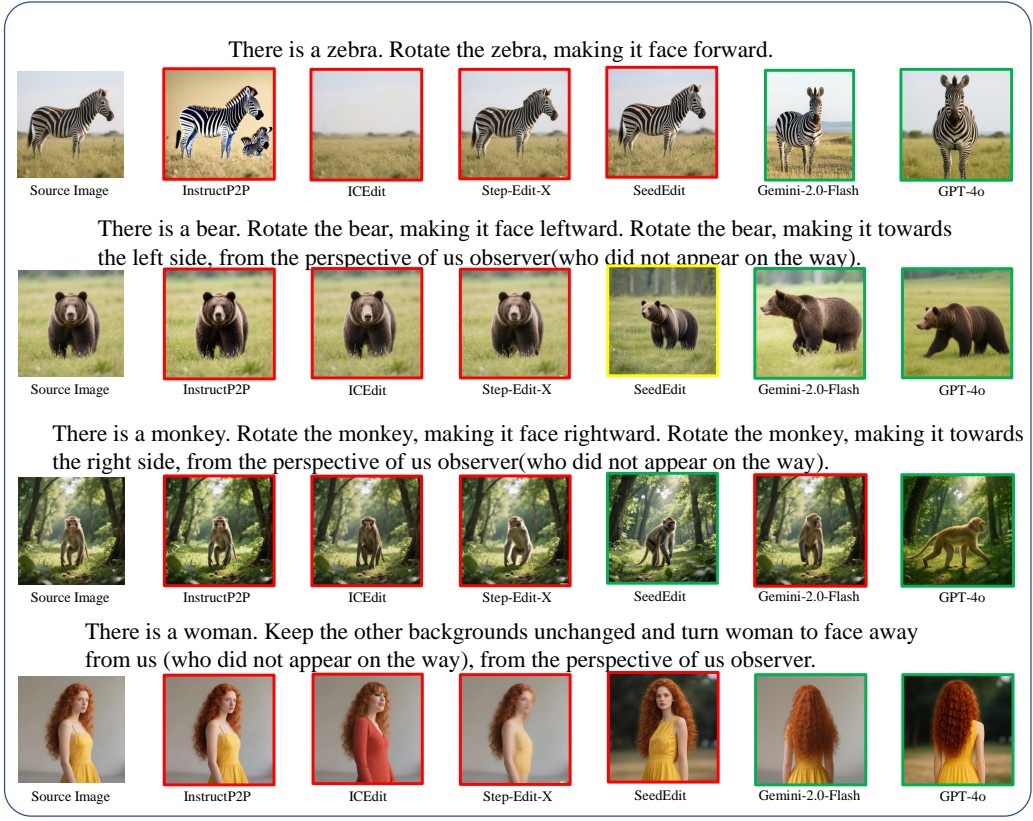

Figure 21: Visualization of Instruction-based Image Editing Benchmark on the subdomain **Object Pose**

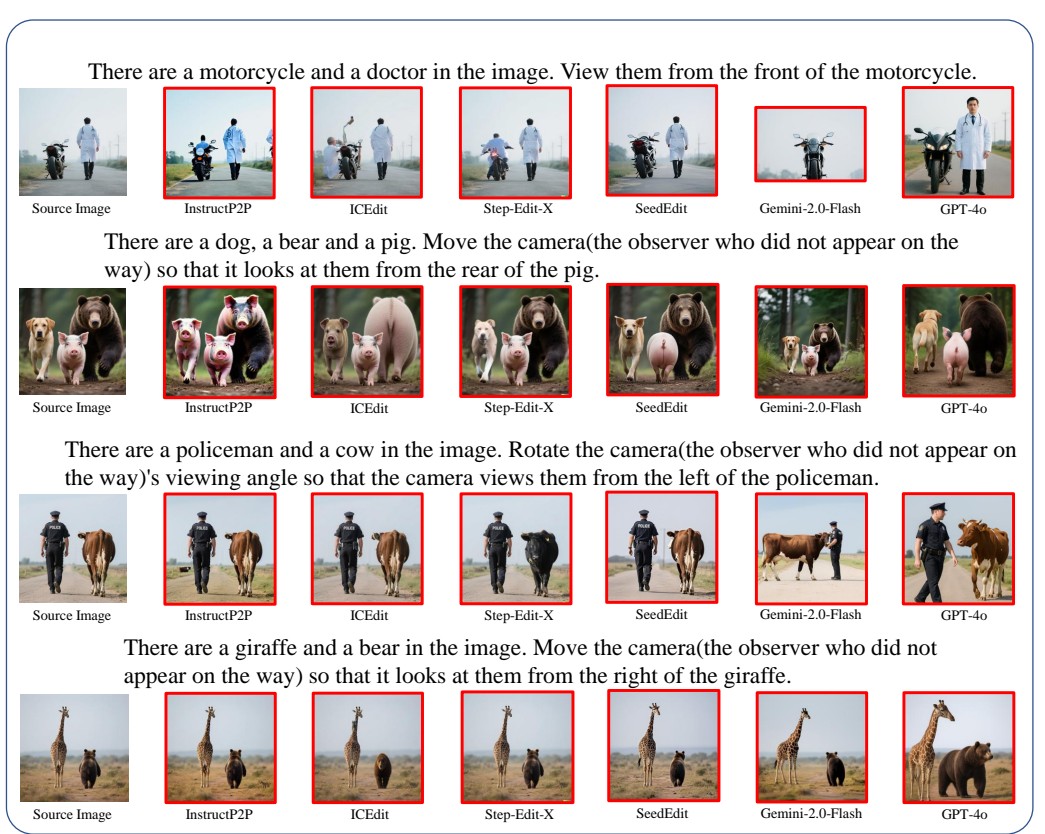

Figure 22: Visualization of Instruction-based Image Editing Benchmark on the subdomain **Complex Pose**

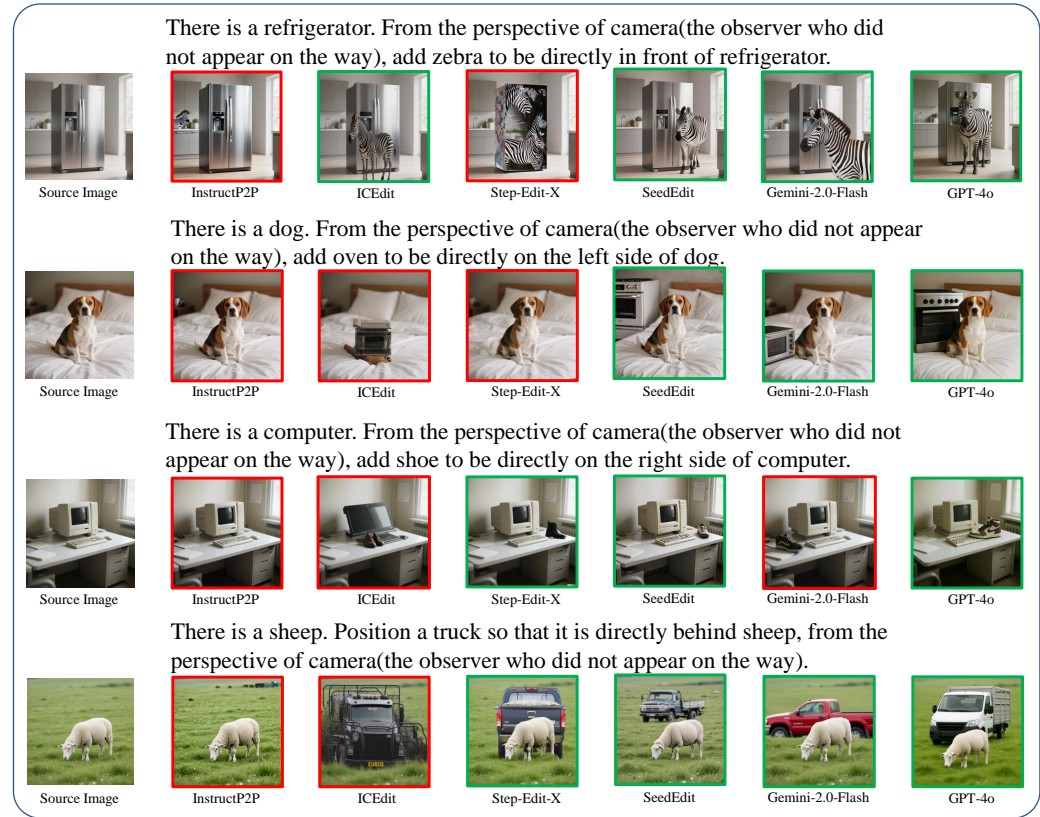

Figure 23: Visualization of Instruction-based Image Editing Benchmark on the subdomain **Egocentric Relation**

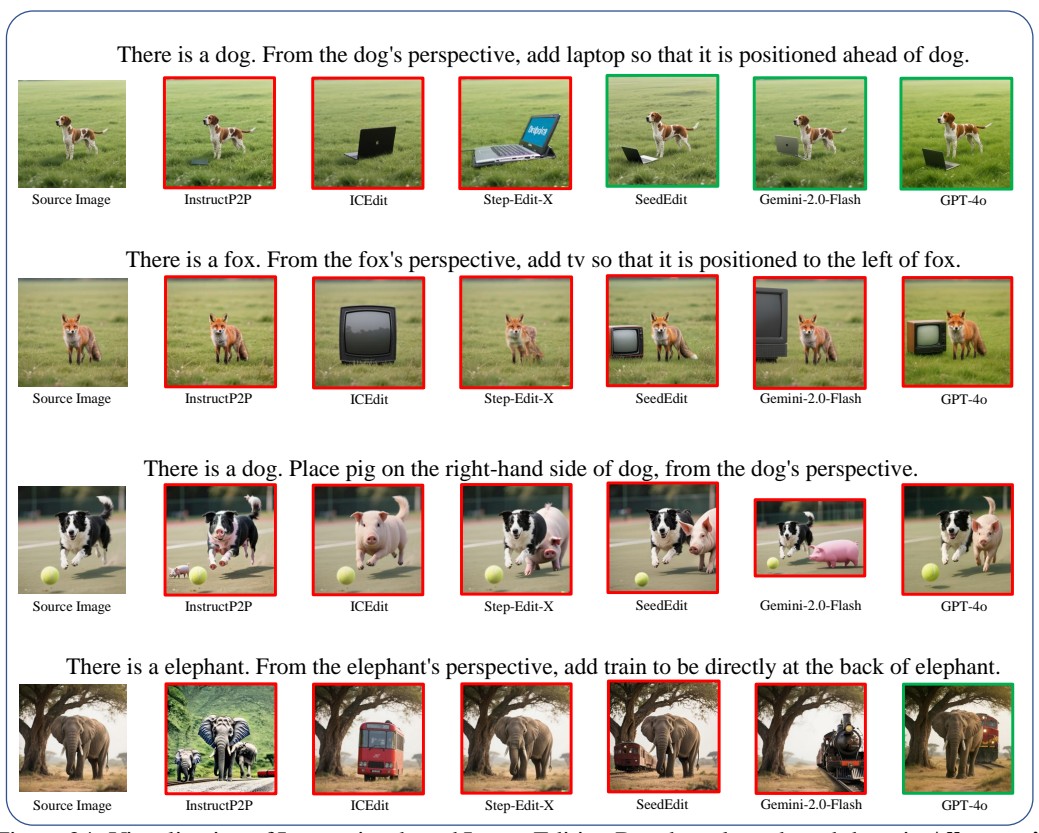

Figure 24: Visualization of Instruction-based Image Editing Benchmark on the subdomain **Allocentric Relation**

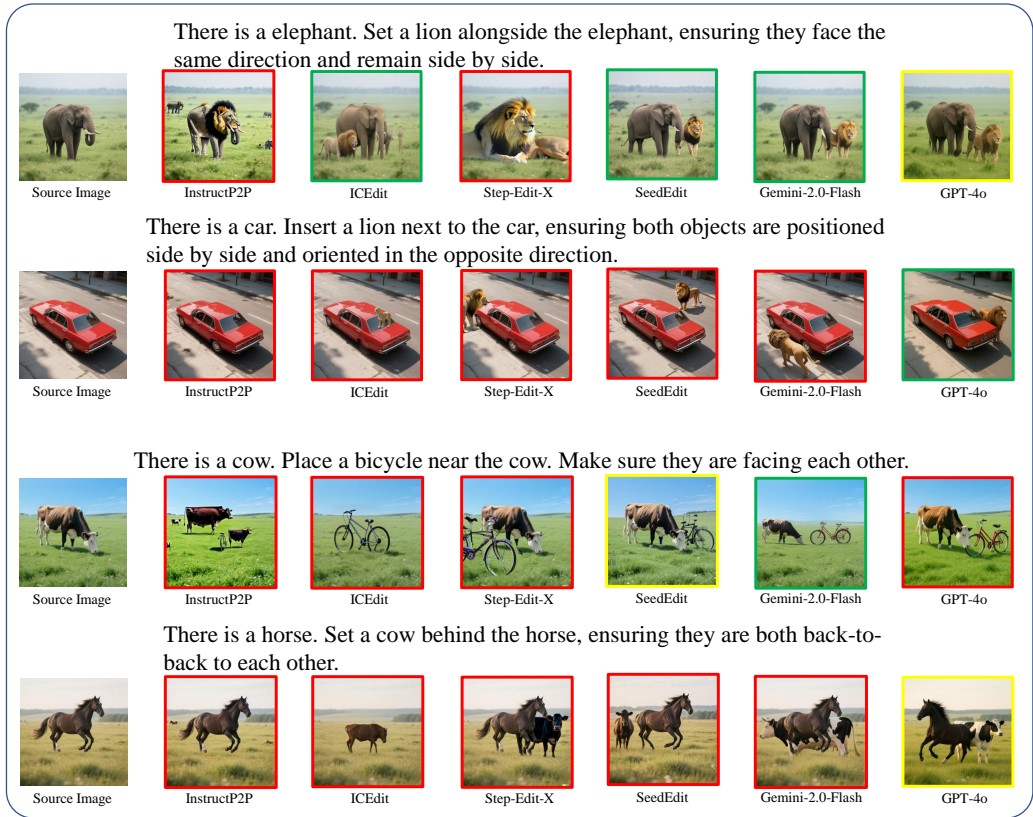

Figure 25: Visualization of Instruction-based Image Editing Benchmark on the subdomain **Intrinsic Relation**

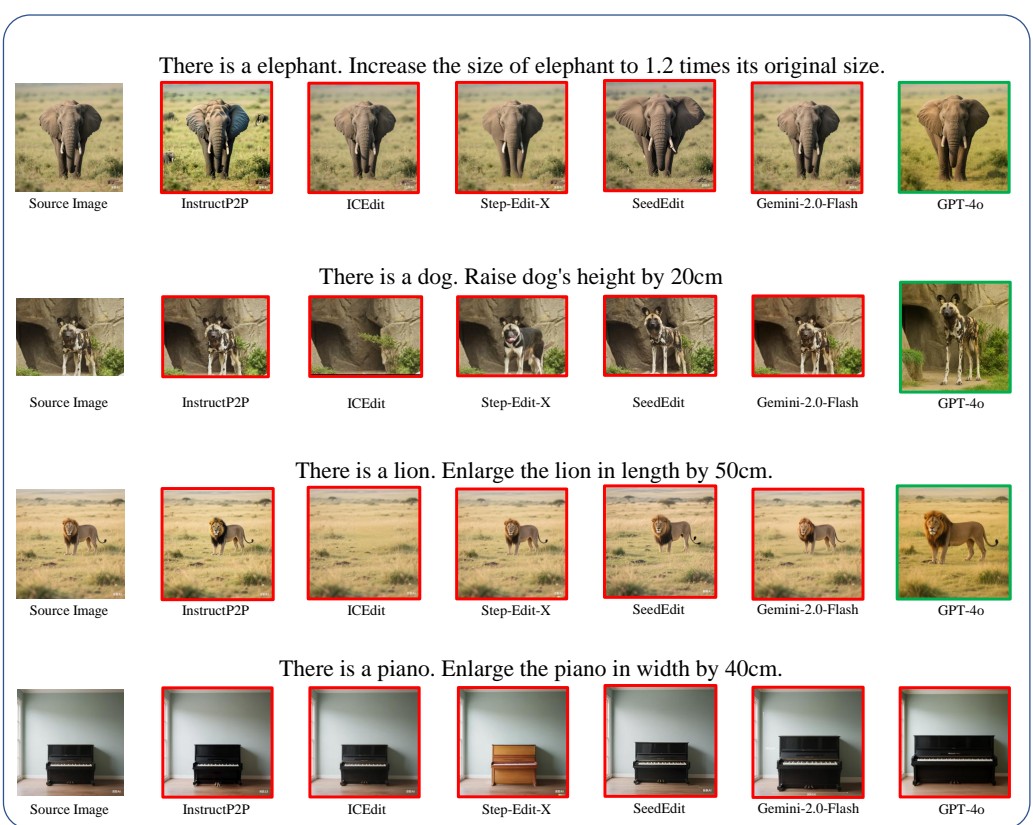

Figure 26: Visualization of Instruction-based Image Editing Benchmark on the subdomain **Object Size**

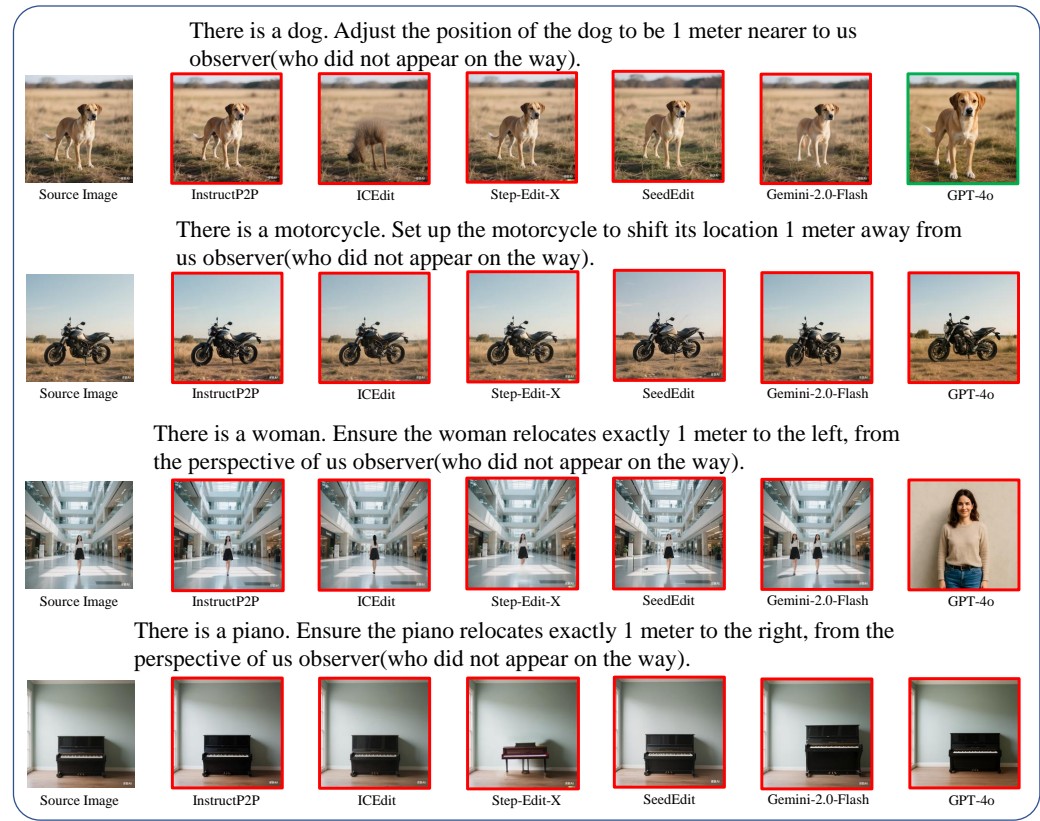

Figure 27: Visualization of Instruction-based Image Editing Benchmark on the subdomain **Object Distance**

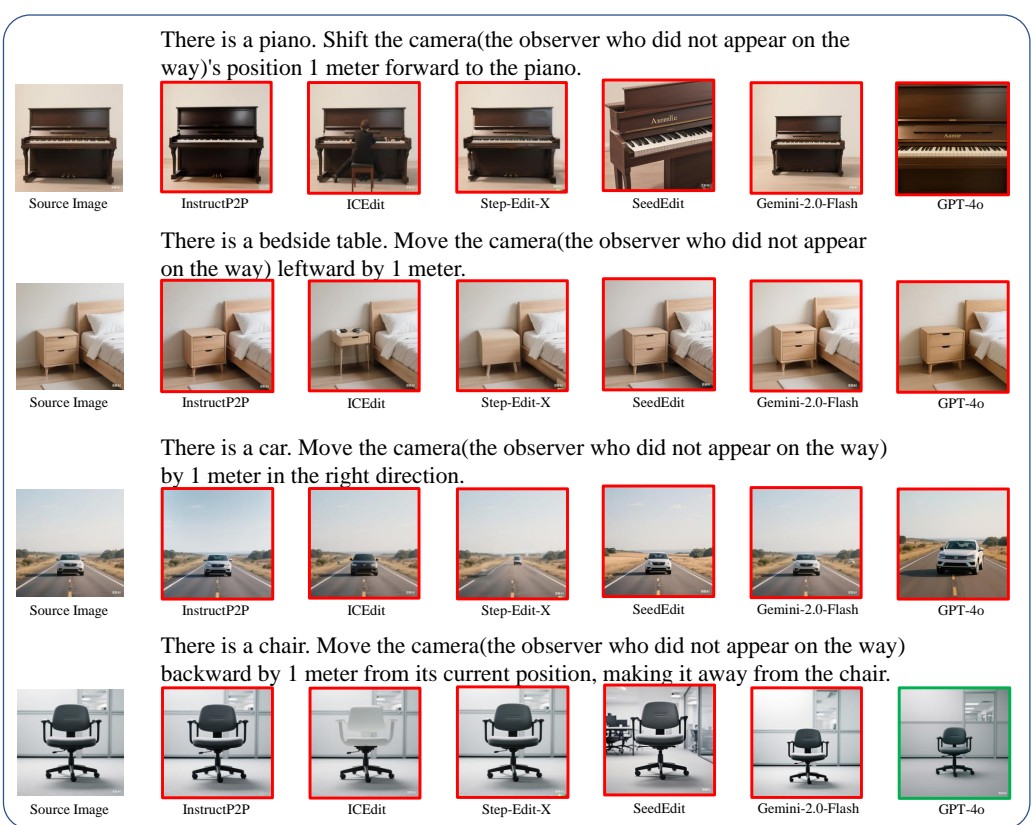

Figure 28: Visualization of Instruction-based Image Editing Benchmark on the subdomain **Camera Distance**

# E  More Result on human alignment of different evaluators

| Evaluator | Spatial Pose | | | Spatial Relation | | | Spatial Measurement | | | Ave. |
|---|---|---|---|---|---|---|---|---|---|---|
| | Camera | Object | Complex | Ego. | Allo. | Intri. | Size | ObjDist | CamDist | |
| qwen-vl-max | 36.0 | 58.0 | 51.0 | 79.0 | 48.0 | 33.0 | 47.0 | 56.0 | 60.0 | 52.00 |
| claude-3-7-sonnet-thinking | 51.0 | 59.0 | 48.0 | 89.0 | 47.0 | 39.0 | 56.0 | 62.0 | 60.0 | 56.78 |

Table 7: Human alignment of different evaluators on spatial understanding, showing their accuracy on manually labeled data.

