# OpenReview forum: "GenSpace: Benchmarking Spatially-Aware Image Generation"
_NeurIPS.cc/2025/Datasets_and_Benchmarks_Track — NeurIPS 2025 Datasets and Benchmarks Track poster_

### Official Review · Reviewer_gPdT · 2025-07-03

**Rating:** 5
**Confidence:** 4

**Summary:**

This paper, "GenSpace," systematically investigates the capability of AI image generation models in adhering to 3D spatial instructions. By constructing a new benchmark, also named GenSpace, the authors test the performance of several mainstream models on tasks involving object pose, spatial relations, and quantitative measurements. The core of this research is a novel evaluation methodology that does not rely on conventional Vision-Language Models (VLMs). Instead, it reconstructs the 3D geometry of the generated images by combining multiple visual foundation models, allowing for a direct, geometric comparison against the prompt. The experimental results demonstrate that current models generally struggle to understand camera perspectives, perform transformations between different frames of reference, and follow specific metric units, revealing a significant gap in the spatial awareness capabilities within this field.

**Additional Feedback:**

This is a solid work that establishes a strong foundation for evaluating spatial intelligence. A highly constructive direction for future work would be to address the uncertainty within the evaluation process. I would suggest exploring methods for the pipeline to output not just a single score, but also a confidence interval. This could be achieved, for instance, by propagating uncertainty from the base models through the pipeline, or by measuring the stability of the score against small input perturbations. This would make the benchmark more robust and fair, especially in cases where the 3D reconstruction itself is ambiguous.

**Dataset Code Accessibility:**

Yes

**Ethical Considerations:**

No, there are no or only very minor ethics concerns

**Final Justification:**

Author's rebuttal resolves my concerns, I'd like keep my rating as Accept

**Limitations Weaknesses:**

* The evaluation method is a multi-stage pipeline (e.g., object detection, then depth estimation). Errors from each stage can accumulate and propagate to the next, potentially leading to significant deviations in the final score. The paper does not fully discuss or quantify the impact of these cascading errors.
* The benchmark primarily uses precise, unambiguous prompts. Consequently, it does not effectively test the model's ability to handle the spatial ambiguity common in natural language (e.g., "near the car"). This is an important challenge for real-world applications.
* Reconstructing a 3D scene from a single 2D image is an inherently ill-posed problem, meaning one 2D image can have multiple valid 3D interpretations. The evaluation pipeline provides a single "most likely" 3D reconstruction. Therefore, using this single reconstruction as the ground truth might unfairly penalize a model that generated a different, yet equally plausible, 3D scene.

**Strengths Contributions:**

* The paper successfully decomposes the abstract concept of "spatial awareness" into three quantifiable dimensions: pose, relation, and measurement. This systematic framework makes the evaluation of model capabilities more structured and targeted.
* Its main contribution is the proposed geometry reconstruction-based evaluation pipeline. This approach bypasses the inherent limitations of existing VLMs in spatial reasoning. By analyzing the 3D structure directly, it offers a more objective, physics-grounded alternative for evaluating such tasks.
* The paper provides robust empirical data from evaluating multiple representative models across both text-to-image and image-editing modes. Its findings, such as the model's bias towards egocentric views and neglect of metric information, are consistent and provide strong evidence about the current limitations of generative models.
* The authors have released the benchmark evaluation code, which is of direct practical value for the community for reproduction, follow-up research, and promoting standardized evaluation in this area.

---

> ### Author Rebuttal · Authors · 2025-07-31
>
> Thank you for your supportive review and suggestions. Below we respond to the comments in **Weaknesses (W)**.
>
> ---
>
> ## W1: Analysis of per-stage error propagation
>
> To more comprehensively and deeply validate and analyze the cumulative error at each step of our evaluation pipeline, we built a rendered test set based on Blender. We manually selected 50 object assets and placed them on a scene plane according to different rules, rendering them from random viewpoints, resulting in 1200 samples in total. Based on the GT object and camera pose, we can easily obtain the qualitative and quantitative spatial relationships among the objects. On this new rendering benchmark, we further tested our evaluation pipeline and analyzed the accumulation of errors in each pre-stage. The results are as follows:
>
> | Object mask | Orientation | Depth      | Camera & Reconstruction | Ave. Score |
> |-------------|-------------|------------|------------------------|------------|
> | predicted   | predicted   | predicted  | predicted              | 76.80      |
> | GT          | predicted   | predicted  | predicted              | 73.69      |
> | GT          | GT          | predicted  | predicted              | 81.62      |
> | GT          | GT          | GT         | predicted              | 90.03      |
> | GT          | GT          | GT         | GT                     | 100.00     |
>
> We found that using predicted object masks yields better results, likely because current detection and segmentation models (Grounding-DINO and SegmentAnything) are very stable. In contrast, the GT masks from rendering often have many small holes, which can affect orientation prediction.
>
> ---
>
> ## W2: Ambiguous spatial description
>
> Thank you for your suggestion. We will try to incorporate ambiguous spatial descriptions into our benchmark in the future. One additional challenge here is that ambiguous descriptions are difficult to score quantitatively. One possible approach is to provide the spatial information extracted by our 3D evaluation pipeline as prior knowledge to a VLM, and let it comprehensively consider object semantics, scene information, and commonsense to provide an overall evaluation.
>
> ---
>
> ## W3: Single reconstruction might be biased
>
> Thank you for your suggestion. Considering the rapid development of visual foundation models, there are constantly new and diverse depth and camera prediction models being proposed. Using different models, such as MoGe v2[1] or DepthPro[2] for reconstruction, may result in multiple reasonable 3D reconstruction results. Ultimately, ensembling these results can provide more stable outcomes, as well as present uncertainty and confidence intervals. Since our method is highly modular, every part of the evaluation pipeline can be easily replaced with other models.
>
> [1] Wang R, Xu S, Dong Y, et al. MoGe-2: Accurate Monocular Geometry with Metric Scale and Sharp Details. arXiv preprint:2507.02546, 2025. \
> [2] Bochkovskii A, Delaunoy A, Germain H, et al. DepthPro: Sharp monocular metric depth in less than a second. (ICLR 2025)
>
> ---

---

> > ### Comment · Reviewer_gPdT · 2025-08-04
> >
> > Thanks authors for the rebuttal, I'd keep my rating as Accept

---

> > > ### Author Response · Authors · 2025-08-05
> > > **Thank you for your support**
> > >
> > > We sincerely appreciate your kind support. In our final revision, we will further enhance the paper by incorporating the valuable insights gained from the rebuttal discussions. Thank you once again for your guidance. Please let us know if you have any further questions or suggestions; we would be pleased to address them.

---

### Official Review · Reviewer_T4TS · 2025-07-03

**Rating:** 5
**Confidence:** 3

**Summary:**

GenSpace introduces a benchmark and evaluation pipeline to assess 3D spatial awareness in image generation models. It defines three dimensions of spatial understanding: **Spatial Pose** (object/camera orientation), **Spatial Relation** (object interactions across perspectives), and **Spatial Measurement** (quantitative control). The benchmark includes 1,800 text-to-image prompts and 1,800 image-editing tasks. The authors propose a novel evaluation pipeline combining visual foundation models (e.g., depth estimation, orientation detection) to reconstruct 3D geometry from 2D images, outperforming general VLMs like GPT-4o in human alignment (76.2% vs. ≤56.4%). Evaluations of 12 models (including SD-XL, DALL-E 3, and GPT-4o) reveal critical limitations in handling allocentric relations, camera positioning, and metric adherence.

**Dataset Code Accessibility:**

Yes

**Ethical Considerations:**

No, there are no or only very minor ethics concerns

**Final Justification:**

Before rebuttal, my major concerns are (1) insufficient dataset size; (2) lack of comparisons with other VLM annotation systems.

In the rebuttal phase, the authors have provided sufficient materials, which include: (1) a comprehensive design to scale up the dataset size; (2) comparison of their scores with Gemini, GPT4o, and GPTo3 scores;

These two additional materials help address my concerns. Considering the fact that dataset scaling is a to-do work, I decided to maintain my score as **5, accept**.

**Limitations Weaknesses:**

1. **Static Scene Focus**: Benchmarks static spatial relationships only; dynamic interactions (e.g., object collisions) are unexplored.
2. **Data Diversity**: Limited to 50 object categories; lacks rare or abstract entities.

**Strengths Contributions:**

1. **Novel Benchmark**: Comprehensive taxonomy covering 9 sub-domains of spatial reasoning across text-to-image and image-editing tasks, grounded in real-world photography principles.
2. **Specialized Evaluation Pipeline**: Addresses limitations of general VLMs by leveraging multi-model 3D reconstruction (depth/orientation/object detection), significantly improving human alignment.
3. **Rigorous Analysis**: Exposes systemic weaknesses in SOTA models (e.g., GPT-4o struggles with allocentric perspectives and metric precision despite high overall capability).
4. **Practical Insights**: Identifies three core limitations: camera localization, egocentric-allocentric transformation, and metric measurement adherence, guiding future research.
5. **Reproducibility**: Open-sourced code/data with detailed documentation.

---

> ### Author Rebuttal · Authors · 2025-07-31
>
> Thank you for your supportive review and suggestions. Below we respond to the comments in **Weaknesses (W)**.
>
> ---
>
> ## W1: Static Scene Focus
>
> Our main goal is to highlight the current limitations of image generation models in understanding basic spatial concepts, as there is still much room for improvement in object pose and basic qualitative/quantitative relationships.
>
> However, as you mentioned, dynamic interaction will be an important future direction. We will try to add test tasks related to dynamic spatial interaction in the future, and use the spatial information extracted by our evaluation pipeline as prefix knowledge for VLMs, allowing VLMs to judge such more ambiguous spatial interactions.
>
> ---
>
> ## W2: Data Diversity
>
> - **For occlusion**, we added additional occlusion sub-tasks based on two templates ("<obj_1> blocks part of <obj_2>" and "<obj_1> is partially blocked by <obj_2>"), with 50 samples for each description. To evaluate whether occlusion exists in the image, we check whether the bounding boxes of the two objects intersect, and if so, use their depth to determine the front-back relationship. Some results of text-to-image generation models on these tasks are as follows:
>
>     | Model         | Occlusion Accuracy |
>     |---------------|-------------------|
>     | SDXL          | 56.67             |
>     | FLUX          | 52.50             |
>     | Seedream-3.0  | 81.25             |
>     | GPT-4o        | 65.00             |
>
> - **For lighting**, we added different lighting conditions in the text prompt (such as "bright lighting" or "dim lighting") over basic spatial pose dimensions. The results are as follows:
>
>     | Model         | Camera | Object | Complex |
>     |---------------|--------|--------|---------|
>     | SDXL          | 30.78  | 26.66  | 2.27    |
>     | FLUX          | 44.16  | 35.48  | 9.99    |
>     | Seedream-3.0  | 46.11  | 45.00  | 11.69   |
>     | GPT-4o        | 54.36  | 61.55  | 20.69   |
>
>     The results show that these additional, spatially irrelevant descriptions do not significantly affect the spatial intelligence of generation models. Moreover, since our evaluation pipeline is based on robust visual foundation models, the evaluation scores are also not significantly affected.
>
> - **For rare objects**, we prompted the LLM to provide ten rare categories ('Ultraman', 'Loom', 'Rickshaw', 'Sedan chair', 'Gargoyle', 'Throne', 'Cowboy', 'Telegraph machine', 'Monster', 'Unicorn'). The results are as follows:
>
>     | Model         | Camera | Object | Complex |
>     |---------------|--------|--------|---------|
>     | SDXL          | 31.26  | 20.81  | 3.48    |
>     | FLUX          | 50.45  | 39.08  | 21.39   |
>     | Seedream-3.0  | 46.66  | 56.62  | 21.49   |
>     | GPT-4o        | 63.82  | 57.69  | 30.69   |
>
>     The results on spatial pose are still similar to those of common categories. This is because the basic capabilities of current generation models have greatly improved, and their ability to generate diverse categories is already very strong. Moreover, the models used in our evaluation pipeline are also well known for their generalization to "anything".
>
> ---

---

> > ### Comment · Reviewer_T4TS · 2025-08-05
> > **Thanks for rebuttal**
> >
> > Thanks for the authors' rebuttal.
> >
> > I think the newly provided materials help strengthen the diversity of the dataset, which I encourage the authors to continue expanding at scale. However, considering that there is still room to enrich the dynamic interaction of the images (as the authors have mentioned), I lean towards maintaining my score as **5, accept**.

---

> > > ### Author Response · Authors · 2025-08-05
> > > **Thank you for your support**
> > >
> > > We sincerely appreciate your kind support. In our final revision, we will further enhance the paper by incorporating the valuable insights gained from the rebuttal discussions. Thank you once again for your guidance. Please let us know if you have any further questions or suggestions; we would be pleased to address them.

---

### Official Review · Reviewer_4WTt · 2025-07-03

**Rating:** 4
**Confidence:** 2

**Summary:**

This paper introduces GenSpace, a novel benchmark and evaluation pipeline designed to assess the spatial reasoning capabilities of modern text-to-image generation and instruction-based image editing models. To evaluate this, the authors define four dimensions of spatial awareness and propose a way based multiple 3D reconstruction models to extract the spatial relationship from 2D images. They conduct experiments to validate the evaluation pipeline with human annotations. The paper concludes by positioning some limitations as key areas for future research to enhance the spatial awareness of generative models.

**Dataset Code Accessibility:**

Yes

**Dataset Code Comments:**

The links for both dataset and code works and the instructions are clear to me.

**Ethical Considerations:**

No, there are no or only very minor ethics concerns

**Final Justification:**

Thanks for the rebuttal. My concerns are partially addressed. I will keep the score as borderline acceptance.

**Limitations Weaknesses:**

1. I am worried about the limited size of human annotations and thus the reliability of alignment results. More stats, e.g., std, PLCC, etc are recommended to further validate the conclusion.
2. The comparison of evaluation pipeline with current VLMs is somehow not enough. There are many alternative ways to evaluate the spatial-relationship except for VLMs. At least, a comparison with 2D only models is advised to validate the rationale of 3D reconstruction.

**Strengths Contributions:**

1. The spatial awareness of generative and editing models is indeed a useful research direction and well-know challenge in the field. Benchmarking this property is meanful.
2. The evaluation protocol of reconstructing 3D scene geometry is reasonable. With 3D reconstructions, spatial relationship can be detected better than only rely on 2D images.

---

> ### Author Rebuttal · Authors · 2025-07-31
>
> Thank you for your supportive review and suggestions. Below we respond to the comments in **Weaknesses (W)**.
>
> ---
>
> ## W1.1: Limited size of human annotations
>
> To more comprehensively and deeply validate and analyze our evaluation pipeline, we built a rendered test set based on Blender. We manually selected 50 object assets and randomly placed them on a scene plane according to different rules, rendering them from random viewpoints, resulting in 1200 samples in total. Based on the GT object and camera pose, we can easily obtain the qualitative and quantitative spatial relationships among the objects. On this new rendering benchmark, we further tested our evaluation pipeline and analyzed the accumulation of errors in each pre-stage. The results are as follows:
>
> | Object mask | Orientation | Depth      | Camera & Reconstruction | Ave. Score |
> |-------------|-------------|------------|------------------------|------------|
> | predicted   | predicted   | predicted  | predicted              | 76.80      |
> | GT          | predicted   | predicted  | predicted              | 73.69      |
> | GT          | GT          | predicted  | predicted              | 81.62      |
> | GT          | GT          | GT         | predicted              | 90.03      |
> | GT          | GT          | GT         | GT                     | 100.00     |
>
> We found that using predicted object masks yields better results, likely because current detection and segmentation models (Grounding-DINO and SegmentAnything) are very stable. In contrast, the GT masks from rendering often have many small holes, which can affect orientation prediction.
>
> ---
>
> ## W1.2: More validation statistics
>
> The additional validation statistics for Table 3 are as follows:
>
> | Evaluator      | Ave.agreement | Ave.PLCC | Ave.std |
> | -------------- | ------- | -------- | --- |
> | Gemini-2.5-pro |     56.44     |       0.280	|0.418   |
> | GPT-4o         |     53.22    |       0.299|	0.381   |
> | GPT-o3         |     55.78    |       0.446	|0.427   |
> | Ours           |     76.22    |       0.768	|0.464    |
>
>
> ---
>
> ## W2: Comparison with 2D only models
>
> In fact, our pipeline is based on the predictions of advanced 2D models. In settings that do not require complex spatial reasoning, such as object pose, we directly rely on the predictions of 2D models for scoring. Therefore, in terms of single abilities (such as depth and orientation recognition), our pipeline is fully consistent with advanced 2D models. The more important significance of our reconstructed scene is that it enables reasoning about more complex relationships in 3D space, which 2D models cannot achieve (such as object relationships and viewpoint understanding).
>
> ---

---

> > ### Comment · Reviewer_4WTt · 2025-08-06
> >
> > Thanks for providing details stats regarding human annotations and validation. My concerns are partially addressed. I will keep the score as borderline acceptance.

---

> > > ### Author Response · Authors · 2025-08-08
> > > **Thank you for your support!**
> > >
> > > We sincerely appreciate your kind support. In our final revision, we will further enhance the paper by incorporating the valuable insights gained from the rebuttal discussions. Thank you once again for your guidance. Please let us know if you have any further questions or suggestions; we would be pleased to address them.

---

### Official Review · Reviewer_CJ2L · 2025-07-03

**Rating:** 4
**Confidence:** 5

**Summary:**

The paper introduces GenSpace, a benchmark aimed at evaluating spatial reasoning in text-to-image generation and instruction-based image editing. It decomposes spatial understanding into nine sub-domains (pose, inter-object relations, metric distance) and scores model outputs via an automatic 3-D analysis pipeline that chains off-the-shelf detectors, depth estimators, and pose/orientation predictors. The benchmark contains 3.6 k synthetic prompts/instructions and reports a 76 % correlation with human ratings on a 900-image subset.

**Additional Feedback:**

Report per-stage error propagation (e.g.\ object detection vs. depth vs. pose) to pinpoint the bottleneck.
Provide a larger, non-templated split using real photographs or Blender scenes to test robustness.
Replace hard thresholds with a learned continuous score (e.g.\ via contrastive regression to human ratings).

**Dataset Code Accessibility:**

Yes

**Dataset Code Comments:**

Partly.
The heavy reliance on third-party checkpoints (Grounded-SAM, Depth-Anything, etc.) means a turnkey Colab is essential for reproducibility; this is not yet provided.

**Ethical Comments:**

No, there are no or only very minor ethics concerns.
Prompts and images are synthetic; no personal data are involved. Bias can still leak from the LLM used to generate templates, but the authors cite an offensive-content filter (Sec. A). I see no major privacy or misuse issues.

**Ethical Considerations:**

No, there are no or only very minor ethics concerns

**Final Justification:**

Thanks for your reply — it really helped clear up my concerns. I’ve raised my score to “borderline accept.”

**Limitations Weaknesses:**

Core detection logic and prompt template strategy are directly adapted from GenEval; the main addition is swapping 2-D IoU for 3-D geometry. This feels more like engineering integration than a new evaluation paradigm.

3.6 k LLM-templated prompts are modest for today’s generative models. Real-world variability (occlusion, lighting, rare objects) is largely absent, limiting external validity.

Key thresholds (e.g.\ 30° pose error, 33 % depth error) are hand-tuned (Sec. 3.3). Pipeline errors compound; the ablation in Table 2 shows depth failure alone flips ~21 % of scores.

Only 900 images are manually rated, and selection criteria are unclear. No analysis is provided on long-tail scenes where the 3-D pipeline is least reliable.

 The pipeline depends on ~5 external models; reproducing results requires matching specific versions and GPU memory footprints that may shift over time.

The paper does not explore how well the score tracks perceptual preferences (e.g.\ does a 10-point drop correspond to a visually obvious error?).

Only 600 instructions target editing; success is assessed on the final image alone, ignoring whether the requested change was performed.

**Strengths Contributions:**

The work targets spatial consistency, a well-known weakness of current T2I systems but under-served by existing benchmarks such as GenEval or T2I-CompBench.

The three-tier, nine-sub-domain hierarchy provides an organized vocabulary for spatial errors and could help future diagnostic studies.

Integrating depth, camera pose, and object pose estimation into a single metric is technically non-trivial and, to my knowledge, new to T2I evaluation.

---

> ### Author Rebuttal · Authors · 2025-07-31
>
> Thank you for your valuable review and suggestions. Below we respond to the comments in **Weaknesses (W)**.
>
> ---
>
> ## W1: Novelty over GenEval
>
> Object detection is only a part of our evaluation pipeline. Our main focus is on the definition and evaluation of spatial intelligence in image generation. Compared to GenEval, which focuses on simple counting and 2D spatial questions (e.g., up, down, left, right), 3D spatial intelligence tasks require much richer spatial perception abilities. To evaluate these abilities, we propose our own insights regarding how to extract spatial information, how to combine and utilize spatial information for various tasks, and how to derive the final scores. We believe these new ideas are meaningful for the community’s understanding of spatial perception in images, and they have not been explored before.
>
> ---
>
> ## W2: Real-world variability
>
> We provide experiments regarding real-world variability:
>
> - **For occlusion**, we added additional occlusion sub-tasks based on two templates ("<obj_1> blocks part of <obj_2>" and "<obj_1> is partially blocked by <obj_2>"), with 50 samples for each description. To evaluate whether occlusion exists in the image, we check whether the bounding boxes of the two objects intersect, and if so, use their depth to determine the front-back relationship. Some results of text-to-image generation models on these tasks are as follows:
>
>     | Model         | Occlusion Accuracy |
>     |---------------|-------------------|
>     | SDXL          | 56.67             |
>     | FLUX          | 52.50             |
>     | Seedream-3.0  | 81.25             |
>     | GPT-4o        | 65.00             |
>
> - **For lighting**, we added different lighting conditions in the text prompt (such as "with bright lighting" or "with dim lighting") over basic spatial pose dimensions. The results are as follows:
>
>     | Model         | Camera | Object | Complex |
>     |---------------|--------|--------|---------|
>     | SDXL          | 30.78  | 26.66  | 2.27    |
>     | FLUX          | 44.16  | 35.48  | 9.99    |
>     | Seedream-3.0  | 46.11  | 45.00  | 11.69   |
>     | GPT-4o        | 54.36  | 61.55  | 20.69   |
>
>     The results show that these additional, spatially irrelevant descriptions do not significantly affect the spatial intelligence of generation models. Moreover, since our evaluation pipeline is based on robust visual foundation models, the evaluation scores are also not significantly affected.
>
> - **For rare objects**, we prompted the LLM to provide ten rare categories ('Ultraman', 'Loom', 'Rickshaw', 'Sedan chair', 'Gargoyle', 'Throne', 'Cowboy', 'Telegraph machine', 'Monster', 'Unicorn'). The results over basic spatial pose dimensions are as follows:
>
>     | Model         | Camera | Object | Complex |
>     |---------------|--------|--------|---------|
>     | SDXL          | 31.26  | 20.81  | 3.48    |
>     | FLUX          | 50.45  | 39.08  | 21.39   |
>     | Seedream-3.0  | 46.66  | 56.62  | 21.49   |
>     | GPT-4o        | 63.82  | 57.69  | 30.69   |
>
>     The results on spatial pose are still similar to those of common categories. This is because the basic capabilities of current generation models have greatly improved, and their ability to generate diverse categories is already very strong. Moreover, the models used in our evaluation pipeline are also well known for their generalization to "anything".
>
> ---
>
> ## W3 & Feedback 2: Hand-tuned thresholds and ablation in Table 2
>
> Thank you for your suggestion. We tried to use learnable thresholds to fit human ratings. The results show that a 21° pose error and 30% distance error achieve the best effect. The average alignment score of the optimal setting is 79.59, while the current hand-tuned thresholds achieve 76.22. This comparable result shows that our pipeline is not sensitive to the choice of threshold.
>
> Regarding "the ablation in Table 2 shows depth failure alone flips ~21% of scores," we could not find any ablation about depth failure in our manuscript. In response to your W4 & Feedback 1, our analysis shows that errors in predicted depth only lead to about 8.4% of score flips.
>
> ---
>
> ## W4 & Feedback 1: Only 900 manually rated images and lack of analysis for 3D pipelines
>
> 1. **Testing data:** To more comprehensively and deeply validate and analyze our evaluation pipeline, we built a rendered test set based on Blender. We manually selected 50 object assets and placed them on a scene plane according to different rules, rendering them from random viewpoints, resulting in 1200 samples in total. Based on the GT object and camera pose, we can easily obtain the qualitative and quantitative spatial relationships among the objects.
>
> 2. **Analysis of the 3D pipeline (per-stage error propagation):** On this new rendering benchmark, we further tested our evaluation pipeline and analyzed the accumulation of errors in each pre-stage. The results are as follows:
>
>     | Object mask | Orientation | Depth      | Camera & Reconstruction | Ave. Score |
>     |-------------|-------------|------------|------------------------|------------|
>     | predicted   | predicted   | predicted  | predicted              | 76.80      |
>     | GT          | predicted   | predicted  | predicted              | 73.69      |
>     | GT          | GT          | predicted  | predicted              | 81.62      |
>     | GT          | GT          | GT         | predicted              | 90.03      |
>     | GT          | GT          | GT         | GT                     | 100.00     |
>
> We found that using predicted object masks yields better results, possibly because state-of-the-art detection and segmentation models (Grounding-DINO and SegmentAnything) are already very robust. In contrast, the GT masks from rendering often have many small holes, which can lead to some prediction errors for the orientation model.
>
> ---
>
> ## W5: Specific versions and GPU memory for our 3D pipeline
>
> We provide the checkpoint of each model for stable reproducibility. Regarding GPU memory, our method performs inference sequentially and does not continuously increase memory usage.
>
> ---
>
> ## W6: How well the score tracks perceptual preferences
>
> We ran the analysis and found that roughly every 20 points corresponds to one visually obvious error. Since we cannot provide more images during the rebuttal phase, more visualizations will be added in the future to illustrate this.
>
> ---
>
> ## W7: Only 600 instructions target editing and only assessed on the final image
>
> First, we actually have 1800 editing instructions, as stated in line 188: "Summing up, this results in 1,800 samples for each task, totaling 3,600 samples across two tasks."
>
> Second, all our editing instructions are about spatial transformations. For the editing task, we judge the spatial state change between the original and the resulting images. In fact, since the original spatial state for each image is known, the final spatial state’s consistency with the expected spatial state is equivalent to whether the editing process matches the requested spatial transformation.
>
> ---

---

> > ### Author Response · Authors · 2025-08-08
> > **Looking forward to your feedback!**
> >
> > Dear Reviewer CJ2L,
> >
> > Thank you for your comments and suggestions, which have been very helpful to us. We have carefully addressed the concerns raised in your reviews and included additional experimental results.
> >
> > As the discussion deadline is approaching, we would greatly appreciate it if you could take some time to provide further feedback on whether our responses adequately address your concerns.
> >
> > Best regards,
> >
> > The Authors

---

### Decision · Program_Chairs · 2025-09-18

**Decision:**

Accept (poster)

**Comment:**

This paper provides a benchmark for evaluating the spatial correctness of generated images, constructing principled templates for the prompts to be evaluated. It also presents a rigorous pipeline for evaluating the spatial correctness of generated images, leveraging image foundation models to extract 3D information. Their experiments expose some issues associated with using current LLMs as evaluators, which illustrates the importance of the work. The main weakness, as stated by the reviewers, is that the pipeline relies on many models and thus may be susceptible to error accumulation; the rebuttal was able to mitigate this issue by quantifying the effect of the error accumulation. Due to the shown significance of the work and the good technical quality, I would recommend to accept.